# Relative Error Fair Clustering in the Weak-Strong Oracle Model

**Vladimir Braverman** [1] **Prathamesh Dharangutte** [2] **Shaofeng H.-C. Jiang** [3] **Hoai-An Nguyen** [4] **Chen Wang** [5,6]
**Yubo Zhang** [3] **Samson Zhou** [6]

## Abstract

We study fair clustering problems in a setting where distance information is obtained from two sources: a strong oracle providing exact distances, but at a high cost, and a weak oracle providing potentially inaccurate distance estimates at a low cost. The goal is to produce a near-optimal *fair* clustering on $n$ input points with a minimum number of strong oracle queries. This models the increasingly common trade-off between accurate but expensive similarity measures (e.g., large-scale embeddings) and cheaper but inaccurate alternatives. The study of fair clustering in the model is motivated by the important quest of achieving fairness with the presence of inaccurate information. We achieve the first $(1+\varepsilon)$-coresets for fair $k$-median clustering using $\text{poly}\left(\frac{k}{\varepsilon} \cdot \log n\right)$ queries to the strong oracle. Furthermore, our results imply coresets for the standard setting (without fairness constraints), and we could in fact obtain $(1+\varepsilon)$-coresets for $(k,z)$-clustering for general $z = O(1)$ with a similar number of strong oracle queries. In contrast, previous results achieved a constant-factor $(> 10)$ approximation for the standard $k$-clustering problems, and no previous work considered the fair $k$-median clustering problem.

The authors are listed in alphabetical order. [1]Google Research [2]Department of Computer Science, Rutgers University [3]School of Computer Science, Peking University [4]Computer Science Department, Carnegie Mellon University [5]Department of Computer Science, Rice University [6]Department of Computer Science & Engineering, Texas A&M University. Correspondence to: Vladimir Braverman <vbraverman@google.com>, Prathamesh Dharangutte <ptd39@rutgers.edu>, Shaofeng Jiang <shaofeng.jiang@pku.edu.cn>, Hoai-An Nguyen <hnnguyen@andrew.cmu.edu>, Chen Wang <chen.wang.research@gmail.com>, Samson Zhou <samsonzhou@gmail.com>.

*Proceedings of the 42nd International Conference on Machine Learning*, Vancouver, Canada. PMLR 267, 2025. Copyright 2025 by the author(s).

## 1 Introduction

Clustering is a cornerstone of unsupervised learning and a fundamental technique for data science, often used to identify latent structure in high-dimensional and large-scale datasets, thus serving as an essential pre-processing step in numerous downstream applications, facilitating dimensionality reduction, feature learning, and data summarization. Informally, the objective of clustering is to partition a dataset into $k$ clusters such that intra-cluster similarity is maximized while inter-cluster similarity is minimized. Traditional clustering formulations, such as the $k$-median and $k$-means problems, have been the focus of extensive study since their inception in the 1950s (Steinhaus et al., 1956; MacQueen, 1967). Formally, in the $k$-median clustering problem, the input is a set $\mathcal{X}$ of $n$ points in a space with metric $\mathbf{d}$, and the goal is to minimize the quantity $\min_{C:|C|\leq k} \sum_{x\in\mathcal{X}} \min_{c\in C} \mathbf{d}(x,c)$.

**Fair clustering.** Observe that the set of centers $C$ inherently divides $\mathcal{X}$ into clusters by assigning each point $x \in \mathcal{X}$ to its closest center $c \in C$. Due to this implicit partitioning rule, standard notions of clustering may produce biased results, leading to potentially discriminatory outcomes that perpetuate social inequalities in downstream applications (Dwork et al., 2012; Chhabra et al., 2021). For instance, clustering algorithms used for hiring or lending may unintentionally discriminate against certain demographic groups, e.g., race, gender, religion, or socioeconomic status. Such biases are particularly concerning in high-stakes domains like healthcare, education, and finance, where data science algorithms are strongly influential in the decision-making process. To address these challenges, there has been a growing body of research focused on incorporating fairness constraints Ahmadian et al. (2020b;a); Song et al. (2024); Chen et al. (2025), especially into clustering algorithms to ensure equitable outcomes.

One such constraint is disparate impact, which requires all "protected" subpopulations to receive similar representation in the decision-making process. To that end, Chierichetti et al. (2017b) introduced $(\alpha, \beta)$-fair clustering, where each point $x \in \mathcal{X}$ is associated with one or more subpopulation (or group) $j \in [\ell]$, across $\ell$ total groups. Since the groups could overlap, we let $\Lambda$ be the maximum number of *disjoint*

*groups* each $x \in \mathcal{X}$ could belong to. The fairness constraints are specified by parameters $0 \leq \alpha_j \leq \beta_j \leq 1$ for each group $j$, so that the objective is to minimize the clustering cost while ensuring that each cluster has at least an $\alpha_j$ fraction and at most a $\beta_j$ fraction of points from group $j$. These constraints are designed to prevent subpopulations from being under-represented or over-represented within any cluster, thereby promoting fair and balanced cluster assignments.

**Weak-strong oracle model.** Up to this point, all formulations of clustering discussed assume that the ground-truth similarity measure between each pair of points is given as part of the input, either through implicitly through an underlying metric or explicitly through either oracle access or a similarity matrix. Recent advancements in modern machine learning have highlighted the importance of large-scale similarity models, particularly in applications involving non-metric data such as images, text, and videos. These models use embedding functions (Van der Maaten & Hinton, 2008; Mikolov et al., 2013; He et al., 2016; Devlin et al., 2019) to provide a mechanism to generate real-valued distances for non-metric data, so that each point $x$ is transformed into a vector $f(x)$ such that the similarity between data points $x$ and $y$ can be inferred from the distance $\mathbf{d}(f(x), f(y))$ in the embedding space. Thus, embedding models, such as those derived from deep learning, have become crucial in enabling downstream tasks such as clustering on non-metric input data.

However, as these embedding models scale in complexity and accuracy, the computational resources required to perform pairwise similarity computations also increase significantly. This has led to the widespread adoption of hybrid approaches that combine highly accurate but resource-intensive similarity models with more efficient, albeit less precise, secondary models (Kusner et al., 2015; John et al., 2020; Silwal et al., 2023; Xu et al., 2024; Galhotra et al., 2024) (see Appendix A for additional discussions). These lightweight models, often referred to as "weak" similarity models, rely on approximations or auxiliary information, such as simple feature-based heuristics (e.g., location, timestamps, or metadata) (John et al., 2020; Ringis et al., 2021), compact neural networks, or historical similarity data (Mitzenmacher & Vassilvitskii, 2022).

To that end, Bateni et al. (2024) introduced the weak-strong oracle model, which considers a space $\mathcal{X}$ consisting of $n = |\mathcal{X}|$ points, equipped with a metric $d : \mathcal{X} \times \mathcal{X} \to \mathbb{R}$ that represents the output of an accurate but computationally expensive similarity model. Although the metric $\mathbf{d}$ is not directly available, it can be accessed through two types of oracles: weak and strong, prescribed as follows.

**Definition 1** (Weak-strong oracle model). Let $(\mathcal{X}, \mathbf{d})$ be a metric space where $d$ is the measure of distance between

points. We assume that the metric $\mathbf{d}$ is *not* directly accessible to the algorithm. In the weak-strong oracle model, we are given two oracles: the weak oracle WO and the strong oracle SO. The properties of the oracles are prescribed as follows.

- Weak oracle WO: given a pair of points $(x, y) \in \mathcal{X} \times \mathcal{X}$, the query $\mathsf{WO}(x, y)$ returns the distance $\mathbf{d}(x, y)$ exactly with probability $2/3$. With probability $1/3$, $\mathsf{WO}(x, y)$ returns an arbitrary answer. The randomness is drawn *exactly once*.

- Strong oracle SO on points: given a pair of points $(x, y) \in \mathcal{X} \times \mathcal{X}$, if both $\mathsf{SO}(x)$ and $\mathsf{SO}(y)$ are queried, then the value of $\mathbf{d}(x, y)$ is (deterministically) revealed.

- Strong oracle SO on distances: given a pair of points $(x, y) \in \mathcal{X} \times \mathcal{X}$, the answer to the query $\mathsf{SO}(x, y)$ (deterministically) reveals $\mathbf{d}(x, y)$.

In this model, the weak oracle distances represent a cost-effective but less accurate approximation of the true distances, whereas the strong oracle provides significantly more precise results at a higher computational cost. Consequently, our objective is to obtain a high-quality solution to clustering for the underlying metric $(\mathcal{X}, \mathbf{d})$, while minimizing the number of queries made to the expensive strong oracle. Furthermore, we allow the corruptions introduced by the weak oracle to be adversarial in nature, which captures a broad class of "imprecise weak oracles", where the weak oracle may produce arbitrarily poor distance estimates with some probability. Since the randomness is drawn *exactly once*, we cannot hope to use repeated queries and majority tricks. We also remark that the failure probability of $\frac{1}{3}$ is quite flexible, e.g., it can be an upper bound on the failure probability across all oracle queries and simply needs to be some constant $< \frac{1}{2}$.

Bateni et al. (2024) achieves constant-factor approximations to $(k, z)$-clustering problems for $z = O(1)$ in the weak-strong oracle model with $\widetilde{O}(k)$ strong point oracle queries and $\widetilde{O}(k^2)$ strong distance oracle queries[1]. They also proved that $\Omega(k)$ strong point oracle queries are necessary for any bounded approximation, making their bounds tight up to polylog $n$ factors. On the other hand, the approximation factor in their algorithm is some $C > 10$ for $k$-means and even bigger constants for general $(k, z)$. Due to the fact that their algorithm is an adaptation of Meyerson sketch (Meyerson, 2001b), any improvement below a certain constant is unlikely. Furthermore, their algorithm does *not* have any guarantees on fairness, which could lead to potentially biased and harmful outcomes. Therefore, there is an open

---

[1] As standard, we use $\widetilde{O}(\cdot)$ to hide polylog $(n)$ terms.

question for efficient clustering algorithms with *fairness constraints* and $(1 + \varepsilon)$-*approximation* in the weak-strong oracle model.

Due to hardness results for $k$-clustering (Guha & Khuller, 1999; Awasthi et al., 2015; Cohen-Addad & Karthik C. S., 2019; Bhattacharya et al., 2021), we cannot hope for *time efficiency* for $(1 + \varepsilon)$-approximate fair clustering. Nevertheless, we could still aim for efficiency on the *query complexity* of the strong oracle. To this end, a very helpful notion is *coresets*. Roughly speaking, a $(k, \varepsilon)$ coreset $\mathcal{S}$ of a point set $\mathcal{X}$ is a weighted subset of points of the $\mathcal{X}$ such that the fair $k$-clustering costs on $\mathcal{S}$ and $\mathcal{X}$ are bounded by a $(1 + \varepsilon)$ factor (see Definition 3 for the formal definition). For our purpose, if we get a coreset of size $s$ with a small number of strong oracle queries, we could then perform clustering as a post-processing step with at most $s$ extra strong point (or $s^2$ distance) oracle queries. As such, we could ask the following motivating question: *Is it possible to design fair $(k, \varepsilon)$ coresets with query efficiency in the weak-strong oracle model?*

### 1.1  Our Contributions

We answer the motivating question in the affirmative by designing the first $(1 + \varepsilon)$ fair $k$-median algorithm with $\text{poly}(k \log n/\varepsilon)$ strong point (and distance) oracle queries and $\text{poly}(k \log n/\varepsilon)$ coreset size. Recall that we use $\Lambda$ to denote the maximum number of disjoint groups a data point belongs to. The formal guarantees of the coreset algorithm can be given as follows.

**Theorem 1.** *There exists an algorithm in the weak-strong oracle model that, given a dataset $\mathcal{X}$ such that each $x \in \mathcal{X}$ belong to at most $\Lambda$ disjoint groups, with high probability computes a $(k, \varepsilon)$ fair coreset of size $\widetilde{O}(\Lambda \cdot \frac{k^2}{\varepsilon^2})$ for $k$-median clustering using $\widetilde{O}(\Lambda \cdot k)$ strong oracle point queries (or edge queries), $\widetilde{O}(\Lambda \cdot nk)$ weak oracle queries, and $\widetilde{O}(\Lambda \cdot (nk + k^2/\varepsilon^2))$ time.*

In addition to the query efficiency on the strong oracle, our algorithm in Theorem 1 also uses the clean uniform sampling framework and enjoys a fast running time, making it suitable for practical purposes. Although getting an *actual* $(1 + \varepsilon)$-approximation clustering might take exponential time, we could run some approximation algorithms, e.g., the algorithms in Chierichetti et al. (2017a); Backurs et al. (2019), as the post-processing to obtain fair clustering. Prior to Theorem 1, no fair $k$-median coreset with non-trivial approximation guarantees or coreset size under the weak-strong oracle model was known[2].

Theorem 1 is stated in the most general form, where the

subpopulation groups could have overlaps. In the case of $\Lambda = 1$, i.e., the subpopulations are themselves disjoint, the construction of fair clustering coreset is essentially reduced to assignment-preserving coreset that specifies the required number of assignments for each cluster (see Section 2 for the formal definition). The following theorem characterizes our guarantees for the assignment-preserving ($\Lambda = 1$ fairness) case.

**Theorem 2.** *There exists an algorithm in the weak-strong oracle model that, given a dataset $\mathcal{X}$, with high probability computes a $(k, \varepsilon)$-coreset of size $O(\frac{k^2 \cdot \log^4 n \cdot \log n/\varepsilon}{\varepsilon^2})$ for assignment-preserving $k$-median clustering using $O(k \log^4 n)$ strong oracle point queries (or edge queries), $O(nk \cdot \log^3 n)$ weak oracle queries, and $\widetilde{O}(nk + k^2/\varepsilon^2)$ time.*

Theorem 2 also implies $(k, \varepsilon)$ coreset for standard $k$-median clustering (without the fairness or assignment constraints). Note that due to the lower bound in Bateni et al. (2024), $\Omega(k)$ strong oracle queries are necessary to obtain such coresets even for any bounded approximation and without assignment constraints. As such, our algorithm is optimal up to $\text{polylog}(n)$ factors.

One would naturally wonder whether we could similarly get $(k, \varepsilon)$ coresets for other $k$-clustering problems, e.g., $k$-means clustering. Our second result shows that such a goal is possible, and if we do *not* care about fairness, we could obtain $(k, \varepsilon)$ coresets for $(k, z)$ clustering with $z = O(1)$.

**Theorem 3.** *There exists an algorithm in the weak-strong oracle model that, with high probability, computes a $(k, \varepsilon)$ coreset of size $\widetilde{O}(\frac{k^2}{\varepsilon^3})$ for $(k, z)$-clustering with any $z = \Theta(1)$ using $\widetilde{O}(\frac{k^2}{\varepsilon^3})$ strong oracle point queries (or edge queries), $\widetilde{O}(nk)$ weak oracle queries, and $\widetilde{O}(nk + k^2/\varepsilon^3))$ time.*

Note that the size and query bounds for $(k, z)$ clustering include $2^{O(z)}$ dependency, although it could be hidden as long as $z = \Theta(1)$. Compared to Theorem 1, the number of strong oracle queries is slightly worse, but we are able to deal with general $(k, z)$ objectives. We leave the quest of the optimal $k$ dependency for $(k, z)$ clustering coreset as an open problem to pursue in the future.

**Experiments.** To validate the theoretical guarantee of our algorithms, we conduct experiments on real-world datasets: the "Adult" dataset (Becker & Kohavi, 1996) and the "Default of Credit Card Clients" dataset (Yeh, 2009). The weak oracle is designed to return very small distances if it runs into an erroneous case. We compare our algorithm against a baseline algorithm that uniformly at random samples and re-weights points. The experiments show that the proposed algorithm in Theorem 1 could significantly outperform the uniform baseline. We defer the experiments to Section 5.

---

[2]For trivial solutions, we can always, e.g., return the entire dataset as the coreset.

## 1.2 Technical overview

Our coreset algorithms are adaptations of the ring sampling approaches used extensively in the literature (e.g., (Chen, 2009; Braverman et al., 2022)). Roughly speaking, the ring sampling approach first performs an $O(1)$-approximation "weak" $k$-clustering algorithm with average cost $R$. Then, for the points assigned to each center, the algorithm divides the points according to *rings* with doubling distances, i.e., $2^i R$ to $2^{i+1} R$ for the $i$-th ring. It has been shown in Chen (2009) that sampling $\widetilde{O}(k/\varepsilon^2)$ points from each ring leads to a $(\varepsilon, k)$-coreset.

In the weak-strong oracle model, a significant challenge for the ring sampling approach is that we do *not* know the set of points in a ring. Since the weak oracle could give arbitrarily adversarial answers, we cannot assign points to rings based on the weak oracle. On the other hand, if we query the strong oracle for all points, the query complexity will be too high to afford.

The key idea to resolve the above dilemma is by *heavy-hitter* sampling and *recursive peeling*. In particular, in each iteration, we sample $k \operatorname{poly}(\log(n)/\varepsilon)$ points, and we make strong oracle queries on the sample points to know their exact ring assignments. We then categorize the rings into the *heavy* ones, i.e., the rings with $k \operatorname{poly}(\log(n)/\varepsilon)$ sampled points, and the *light* one, i.e., the rings with few samples. The key observation here is that if a heavy ring lies on the outside of (possibly multiple) light rings, then with high probability, we could safely ignore the light rings since their costs are negligible compared to the cost of the outer heavy ring. Therefore, we could focus on building coresets only for the heavy rings. Furthermore, after one iteration, we could use a procedure in Bateni et al. (2024) to *peel off* the points in the heavy ring we processed along with the light rings inside. We could then recursively process all the points with a low overhead of iterations.

The above gives the idea of the *unconstrained* $(\varepsilon, k)$-coreset. To generalize the idea to assignment-preserving and fair coresets, we need to adapt the ideas in existing work for fair clustering (e.g., (Cohen-Addad & Li, 2019; Braverman et al., 2022)) in a *white-box* manner. Algorithmically, we introduce an additional *sampling step* from the points we peel off in each iteration. If the peeled points form a ring, we could use lemmas from Cohen-Addad & Li (2019) to establish the desired results. However, the peeled points are not necessarily rings in our algorithm; and in light of this, we need to open the black-box and prove the assignment-preserving properties still hold in our case, which is part of our technical contributions.

To gradually build up the necessary technical tools and for the ease to understand of the readers, we present the results in a *reverse* order. We will first discuss the *unconstrained*

coreset algorithm in Section 3 for $k$-median, which includes the key ideas used in the algorithms. We will then discuss the fair coreset in Section 4.

## 2 Preliminaries

**$k$-clustering and coresets.** We define the distance between a point $x \in \mathcal{X}$ and a set of points $S \subseteq \mathcal{X}$ as $\mathbf{d}(S, x) = \min_{s \in S} \mathbf{d}(s, x)$, or the distance between $x$ and the closest point in $S$. We define the distance between two sets $S_1, S_2 \subseteq \mathcal{X}$ as $\mathbf{d}(S_1, S_2) = \min_{s_2 \in S} \mathbf{d}(S_1, s_2)$. Additionally, we say that the diameter of a set $\operatorname{diam}(S \subseteq \mathcal{X}) = \max_{s_1, s_2 \in S} \mathbf{d}(s_1, s_2)$. We assume the distances are *integers*, and the aspect ratio $\Delta = \operatorname{poly}(n)$: both assumptions are common in the literature.

Let $\mathcal{X}$ be a set of $n$ points. Every point $x \in \mathcal{X}$ has an associated positive integer weight $\mathbf{w}(x)$. Note that having an unweighted set $\mathcal{X}$ is the same as all points being assigned unit weight. We define the total weight of $\mathbf{w}(\mathcal{X}) = \sum_{x \in \mathcal{X}} \mathbf{w}(x)$. We denote a clustering of set $\mathcal{X}$ by center set $\mathcal{C} = \{c_1, \dots, c_k\} \subseteq \mathcal{X}$ as a partition where each $x \in \mathcal{X}$ is assigned to the closest $c \in \mathcal{C}$.

**Definition 2** $((k, z)$-clustering, $k$-means, and $k$-median). We denote the cost of the $(k, z)$-clustering of point $x \in \mathcal{X}$ using center set $\mathcal{C}$ as $\operatorname{cost}_z(\mathcal{C}, x) = \mathbf{d}^z(\mathcal{C}, x)\mathbf{w}(x)$. The overall cost of the $(k, z)$-clustering is defined as

$$\operatorname{cost}_z(\mathcal{C}, \mathcal{X}) = \sum_{x \in \mathcal{X}} \operatorname{cost}_z(\mathcal{C}, x).$$

For $z = 1$, the objective is the $k$-median clustering; for $z = 2$ problem, the objective is called the $k$-means clustering.

Throughout, we use $\mathcal{C}^*$ to denote the *optimal* clustering that minimizes the cost. We now define the notion of $(k, \varepsilon)$ coreset formally for clustering objectives.

**Definition 3** $((k, \varepsilon)$-coreset). Given a set of points $\mathcal{X}$, a weighted subset $S \subseteq \mathcal{X}$ is a $(k, \varepsilon)$-coreset for $\mathcal{X}$ for $(k, z)$-clustering if we have

$$|\operatorname{cost}_z(\mathcal{C}, S) - \operatorname{cost}_z(\mathcal{C}, \mathcal{X})| \leq \varepsilon \cdot \operatorname{cost}_z(\mathcal{C}, \mathcal{X})$$

for *all* $\mathcal{C} \subseteq \mathcal{X}$ satisfying $|\mathcal{C}| \leq k$.

**Fair $k$-clustering.** We formally introduce the notions of *fair clustering* and *assignment-preserving coresets* that are essential for fair coresets. To this end, we first define the notion of *assignment function* and *assignment constraints* in clustering.

**Definition 4** (Assignment constraints in clustering). For any fixed weighted set $S \subseteq \mathcal{X}$ and $\mathcal{C} \subseteq \mathcal{X}$, an assignment constraint is a function $\Gamma : \mathcal{C} \to \mathbb{R}^{\geq 0}$ such that $\sum_{c \in \mathcal{C}} \Gamma(c) = \mathbf{w}(S)$. An assignment function $\sigma : S \times \mathcal{C} \to \mathbb{R}^{\geq 0}$ is said to be consistent with $\Gamma$, denoted as $\sigma \sim \Gamma$, if $\forall c \in \mathcal{C}$,

$\sigma(S, c) := \sum_{x \in S} \sigma(x, c) = \Gamma(c)$. For $S_1 \subseteq S$ and $\mathcal{C}_1 \subseteq \mathcal{C}$, the connection cost between $S_1$ and $\mathcal{C}_1$ under $\sigma$ is defined as:

$$\text{cost}_z^\sigma(S_1, \mathcal{C}_1) := \sum_{x \in S_1} \sum_{c \in \mathcal{C}_1} \sigma(x, c) \cdot \mathbf{d}^z(x, c).$$

The notion of fair $(k, z)$-clustering with group fairness is defined using assignment constraints. Conceptually, fairness here means that the assignment from each group to each cluster satisfies a certain range of group disparity constraints.

**Definition 5.** In $(\alpha, \beta)$-fair $(k, z)$ clustering, the input dataset $\mathcal{X}$ is partitioned into $\mathcal{X}_1, \cdots \mathcal{X}_m$ groups. The groups are not necessarily disjoint. We are further given $m$-dimensional vectors $\alpha, \beta \in [0, 1]^m$. The goal is to find an assignment function $\sigma$ and a set of centers $\mathcal{C} \subseteq \mathcal{X}$ such that for every group $\mathcal{X}_i$ and center $c \in \mathcal{C}$, there is

$$\alpha_i \leq \frac{\sigma(\mathcal{X}_i, c)}{\sigma(\mathcal{X}, c)} \leq \beta_i.$$

**Assignment-preserving clustering and fair $k$-clustering.** Next, we define $(k, z)$-clustering with *assignment constraints*, which, roughly speaking, specifies the "amount" of assignment each center could get. We will see shortly that their is a strong relationship between clustering with assignment constraints and fair clustering.

**Definition 6** ($(k, z)$-Clustering with assignment constraints). Given a weighted dataset $S \subseteq \mathcal{X}$, a center set $\mathcal{C} \subseteq \mathcal{X}$ with $|\mathcal{C}| \leq k$, and an assignment constraint $\Gamma : \mathcal{C} \to \mathbb{R}^{\geq 0}$, the objective for $(k, z)$-Clustering with assignment constraint $\Gamma$ is defined as:

$$\text{cost}_z(S, \mathcal{C}, \Gamma) := \min_{\sigma: S \times \mathcal{C} \to \mathbb{R}^+, \sigma \sim \Gamma} \text{cost}_\sigma^z(S, \mathcal{C}).$$

The next notion introduces the *assignment-preserving* $(k, \varepsilon)$ coresets. It is in the same spirit of $(k, \varepsilon)$-coreset as in Definition 3 but with assignment-preserving properties.

**Definition 7** (Assignment-preserving coresets for $(k, z)$-Clustering). Let $\mathcal{X}$ be a (potentially weighted) dataset. A weighted subset $S \subseteq \mathcal{X}$ is an assignment-preserving $(k, \epsilon)$-coreset for $(k, z)$-clustering if $\mathbf{w}(\mathcal{X}) = \mathbf{w}(S)$, and for every $\mathcal{C} \subseteq \mathcal{X}$ with $|\mathcal{C}| \leq k$ and assignment constraint $\Gamma : \mathcal{C} \to \mathbb{R}^{\geq 0}$:

$$|\text{cost}_z(S, \mathcal{C}, \Gamma) - \text{cost}_z(\mathcal{X}, \mathcal{C}, \Gamma)| \leq \varepsilon \cdot \text{cost}_z(\mathcal{X}, \mathcal{C}, \Gamma).$$

Note that the actual assignment function $\sigma$ that attains $\text{cost}_z(\mathcal{X}, \mathcal{C}, \Gamma)$ and $\sigma'$ that attains $\text{cost}_z(S, \mathcal{C}, \Gamma)$ could be different as they may have different input spaces to begin with.

There is a strong relationship between assignment-preserving coresets and fair coresets. In particular, we have the following established result.

**Proposition 1** ((Huang et al., 2019)). *Let $\mathcal{A}$ be an algorithm that constructs an assignment-preserving $(k, \varepsilon)$ coreset for $(k, z)$-clustering on $\mathcal{X}$ with size $s$. Furthermore, let $\Delta$ be the number of distinct groups each $x \in \mathcal{X}$ could belong to. Then, the following algorithm*

- *Computes $(k, \varepsilon)$ coreset for $(k, z)$-clustering for each distinct group;*

- *Take the union of the coresets;*

*is an algorithm for $(\alpha, \beta)$-fair $(k, \varepsilon)$ coreset for $(k, z)$-clustering.*

Proposition 1 implies the following lemma for the fair coreset in the weak-strong oracle model.

**Lemma 2.1.** *For a given dataset $\mathcal{X}$, let $\Lambda$ be the number of distinct groups each $x \in \mathcal{X}$ could belong to. An algorithm that constructs an assignment-preserving $(k, \varepsilon)$ coreset for $(k, z)$-clustering with size $s_1$, $s_2$ strong oracle queries, $s_3$ weak oracle queries, and time $t$ implies an algorithm for $(k, \varepsilon)$ fair coreset for $(k, z)$-clustering with size $O(\Lambda \cdot s_1)$, $O(\Lambda \cdot s_2)$ strong oracle queries, $O(\Lambda \cdot s_3)$ weak oracle queries, and time $O(\Lambda \cdot t)$*

**Weak-strong oracle primitives.** We now discuss some primitive algorithms in the weak-strong oracle model. We first give a one-way reduction from SO on distances to SO on points.

**Observation 1.** *Given a set of points $(\mathcal{X}, d)$ in a metric space $d$ and any fixed function $f(\mathcal{X}, d)$, if there exists an algorithm that computes $f(\mathcal{X}, d)$ with $q$ strong oracle queries on* points*, then there exists an algorithm that computes $f(\mathcal{X}, d)$ with $O(q^2)$ strong oracle queries on* distances*.*

Note that the other way of reduction (points to distance) does *not* necessarily hold as in Observation 1. Therefore, up to a quadratic blow-up, our primary focus is the *point queries* for the strong oracle SO[3].

Next, we give some existing results from Bateni et al. (2024) on which we build our algorithms. The first technical tool from Bateni et al. (2024) is an algorithm to construct *weak coresets*.

**Proposition 2** ((Bateni et al., 2024), Theorem 8). *There exists an algorithm that given a set of points $\mathcal{X}$ such that $|\mathcal{X}| = n$, returns an $O(1)$-approximate solution to the optimal $(k, z)$-clustering for $z = O(1)$ on $\mathcal{X}$ together with the assignment of each point using $O(k \cdot \log^2 n)$ strong oracle queries and $O(nk \cdot \log^2 n)$ weak oracle queries.*

---

[3]That being said, coreset algorithms actually enjoy the same asymptotic query efficiency for strong point and distance oracles due to algorithm design.

Next, we give a *distance estimator* that is helpful for using weak oracle queries to compute *approximate* distances.

**Proposition 3** ((Bateni et al., 2024), Lemma 6)**.** *Let $S$ be a set of points such that $|S| \geq 10 \log n$, and suppose that strong oracle queries have been made on all points $s \in S$. Furthermore, let $R_S$ be the diameter of $S$. Then, there exists an algorithm that given $S$, for any $x \in \mathcal{X}$ and $s \in S$, uses $\widetilde{O}(1)$ weak oracle queries and returns a estimated distance $\widetilde{\mathbf{d}}(x, S)$ such that*

$$\mathbf{d}(x, s) \leq \widetilde{\mathbf{d}}(x, S) \leq \mathbf{d}(x, s) + C_0 \cdot R_S,$$

*where $C_0 \leq 5$ is a universal constant.*

**Ring sampling primitives** We use the idea of uniform sampling from *rings* in the same manner of Chen (2009); Braverman et al. (2022). In what follows, we introduce the following definition for ring sampling to introduce our algorithm.

**Definition 8.** Let $\nu(A, X) \leq \beta \nu_{\text{OPT}}$ denote the clustering cost and let $R = \frac{\nu(A,X)}{\beta n}$ be the lower bound on average radius of optimal clustering. Let $P_i$ be the set of points $x \in \mathcal{X}$ assigned to $c_i$ and let $P_{i,j} = P_i \cap [B(c_i, (2C_0)^j R) \setminus B(c_i, (2C_0)^{j-1} R)]$ for $j \geq 1$ and $P_{i,j} = P_i \cap B(c_i, R)$ for $j = 0$. As a result, we have $\log(\beta n)$ rings around each $c_i \in \mathcal{C}$ and $x \in \mathcal{X} \setminus \mathcal{C}$ is part of exactly one ring.

Note that compared to the standard ring sampling-based algorithms like Chen (2009), our algorithm uses search with $C_0$ multiplication. This ensures whenever a point is placed in the wrong ring, it could only end up in the next ring.

**The aspect ratio and the number of rings.** Since the rings are defined by a multiplicative factor of $C_0 > 1$, for instances with an aspect ratio $\Delta$, for each center $c_i$ of the $O(1)$-approximation solution, there is at most $O(\log_{C_0}(\Delta))$ rings. It is common to assume the aspect ratio $\Delta = \text{poly}(n)$; for the clarity of presentation, we assume there are exactly $\log n$ rings in this paper. Any change within the regime of polynomial aspect ratio will only result in a constant factor difference in the coreset size.

## 3 A $(1 + \varepsilon)$-Coreset for $k$-median Clustering

In this section, we give our algorithm that produces a $(1+\varepsilon)$-coreset (without fairness constraints) for $(k, z)$-clustering with $\widetilde{O}(k^2/\varepsilon^3)$ points and strong oracle SO queries. We first remind the readers of the guarantees in our theorem. For ease of presentation, we focus on the $k$-median case for the analysis, and defer the full proof of $(k, z)$-clustering to Appendix D. The theorem we aim to prove in this section is as follows.

**Theorem 4.** *There exists an algorithm in the weak-strong oracle model that, with high probability, computes a $(k, \varepsilon)$*

*coreset of size $\widetilde{O}(\frac{k^2}{\varepsilon^3})$ for $k$-median clustering using $\widetilde{O}(\frac{k^2}{\varepsilon^3})$ strong oracle point queries (or edge queries), $\widetilde{O}(nk)$ weak oracle queries, and $\widetilde{O}(nk)$ time.*

**The algorithm.** The main idea of our algorithm is based on the *heavy-hitter sampling* and *recursive peeling*. Roughly speaking, our algorithm first conducts uniform sampling for the points assigned to a center in the $O(1)$-approximation. The algorithm then makes strong oracle queries on these points (or strong distance oracle queries between these points and the center) to determine the rings for the sampled point. Crucially, this allows us to identify the *heavy-hitter ring* that accounts for a significant portion of the cost. As such, it suffices to only construct coresets for these heavy-hitter rings, and we indeed would have enough samples from such rings. At this point, we would argue that there is a collection of rings whose points are either added to the coreset or their costs are insignificant enough so that they could be ignored. We will then "peel" these points from the point set and recurse on the remaining set. The detailed description of the algorithm is as follows.

---

**Algorithm 1. The construction algorithm for $(1 + \varepsilon)$ coreset**

(1) Run the algorithm of Proposition 2 to get the weak approximate clustering WC.

(2) Initialize all rings $P_{i,j}$ as "*not processed*".

(3) Initialize the strong coreset SC $\leftarrow \emptyset$.

(4) For each center $c_i$ of WC:

  (a) For each ring $P_{i,j}$, let $S_{i,j}$ be some samples from the ring specified later.

  (b) Initialize $\widetilde{P}_i \leftarrow P_i$ as the remaining number of points.

  (c) For $r = 1 : 10 \log^2 n$ iterations:

    i. Sample $s_r = 100C_0 \cdot \frac{k \log^3 n}{\varepsilon^3}$ points uniformly at random from $\widetilde{P}_i$, and let the sample set be $S_r$.

    ii. Make strong oracle SO queries as follows:

     • For point SO queries, query $\mathsf{SO}(x)$ for all points $x \in S_r$.

     • For distance SO queries, query $\mathsf{SO}(x, c_i)$ for all points $x \in S_r$.

    iii. **Ring assignment:** For each point $x \in S_r$, add $x$ to $S_{i,j}$, where $j$ is the index of the ring to which $x$ belong.

    iv. Find the ring $j^*$ with the *largest index* such that $|S_{i,j^*}| \geq 80C_0 \cdot \frac{k \log^2 n}{\varepsilon^3}$.

    v. For each ring $\ell$ with index $\ell \leq (j^* + 1)$ and $k$ being "*not processed*":

---

- If $|S_{i,\ell}| \geq 30 \cdot \frac{k \log^2 n}{\varepsilon^2}$, run CORESET-UPDATE (Algorithm 2) with inputs $S_{i,\ell}$ and SC to obtain an updated SC.

vi. Conduct the **peeling** step based as follows.

- If $j^* \neq 0$, run PEELING (Algorithm 3) with $\{S_{i,j}\}_{j=1}^{j^*}$ and $\widetilde{P}_i$.
- Otherwise (if $j^* = 0$):
  ○ If more than $\frac{|S_{i,0}|}{2}$ points has distance at most $R/2C_0$ to $c_i$, then run CONSERVATIVE-PEELING (Algorithm 4) with $S_{i,0}$ and $\widetilde{P}_i$.
  ○ Otherwise, run PEELING (Algorithm 3) with $\{S_{i,0}\}$ and $\widetilde{P}_i$.

vii. Mark rings as "*processed*" with the following rule:

- If CONSERVATIVE-PEELING (Algorithm 4) is executed (which implies $j^* = 0$), mark points in $P_{i,0}$ as "*processed*".
- Otherwise, mark all the points in rings $P_{i,\ell}$ for $\ell \leq j^* + 1$ as "*processed*".

---

**Algorithm 2** (CORESET-UPDATE). **An algorithm that adds points to coreset.**
**Inputs: A sampled point set $S_{i,\ell}$; a set of existing coreset SC**
**Output: An updated set of coreset SC**

(1) Arbitrarily split the set of sampled points $S_{i,\ell}$ to

- $S_{i,\ell}^{\text{est}}$ with $1/3$ of the points in $S_{i,\ell}$;
- $S_{i,\ell}^{\text{weight}}$ with $2/3$ of the points in $S_{i,\ell}$.

(2) Let $\widetilde{m}_{i,\ell} \leftarrow \frac{3|\widetilde{P}_i|}{s_r} \cdot \left|S_{i,\ell}^{\text{est}}\right|$ be the estimated number of points in $P_{i,j}$.

(3) Re-weighting: add each point $x \in S_{i,\ell}$ to the coreset SC with weight $\frac{\widetilde{m}_{i,\ell}}{|S_{i,\ell}|}$.

---

**Algorithm 3** (PEELING). **An algorithm that removes points.**
**Inputs: Sampled sets $\{S_{i,j}\}_{j=1}^{j^*+1}$, where $j^*$ is obtained from line 4(c)iv; the current set of surviving points $\widetilde{P}_i$**
**Output: An updated set of surviving points $\widetilde{P}_i$**

(1) For each point $x \in \widetilde{P}_i$, run the algorithm of Proposition 3 with the set $S^{\text{peel}}$ as an arbitrary subset of $10 \log n$ points in the subset of $\cup_{j=1}^{j^*} S_{i,j} \cup \{c_i\}$.

---

(2) Let $T_i^{(r)}$ be the set of points such that $\widetilde{\mathbf{d}}(x, S^{\text{peel}}) \leq (2C_0)^{j^*+1} \cdot R$.

(3) Remove all points in $T_i^{(r)}$ from $\widetilde{P}_i$, i.e., $\widetilde{P}_i \leftarrow \widetilde{P}_i \setminus T_i^{(r)}$.

---

**Algorithm 4** (CONSERVATIVE-PEELING). **An algorithm that removes points.**
**Inputs: Sampled sets $\{S_{i,0}\}$; the current set of surviving points $\widetilde{P}_i$**
**Output: An updated set of surviving points $\widetilde{P}_i$**

(1) For each point $x \in \widetilde{P}_i$, run the algorithm of Proposition 3 with the set $S^{\text{peel}}$ as an arbitrary subset of $10 \log n$ points in the subset of $S_{i,0} \cup \{c_i\}$.

(2) Let $T_i^{(r)}$ be the set of points such that $\widetilde{\mathbf{d}}(x, S^{\text{peel}}) \leq R$.

(3) Remove all points in $T_i^{(r)}$ from $\widetilde{P}_i$, i.e., $\widetilde{P}_i \leftarrow \widetilde{P}_i \setminus T_i^{(r)}$.

---

**The analysis.** Due to space limits, we only prove the query efficiency and coreset size, and defer the correctness proof to Appendix B.

**Lemma 3.1.** *Algorithm 1 uses at most $k^2 \cdot \frac{\log^5 n}{\varepsilon^3}$ SO point (or distance) queries and at most $O(nk \log^3 n)$ weak oracle queries. The algorithm converges in $\widetilde{O}(nk + \frac{k^2}{\varepsilon^3})$ time.*

*Proof.* By Proposition 2, the number of strong oracle queries required to get WC is $O(k \log^2 n)$ strong point SO queries or $O(k \log^2 n)$ strong distance queries. In addition, for each center for $O(\log^2 n)$ iterations, the algorithm samples $O(k \log^3 n/\varepsilon^3)$ points and makes strong oracle queries on all of then (or the queries between them and the center $c_i$). Since there are at most $k$ centers in WC, the number of strong oracle queries here is $O(k^2 \log^5 n/\varepsilon^3)$.

For the number of weak oracle queries, the WC requires $O(nk \log^2 n)$ weak oracle queries. Furthermore, each point makes at most $O(\log n)$ queries during one iteration of line 4c in Algorithm 1. There are at most $k$ centers, and each center induces $O(\log^2 n)$ iteration, which lead to a total of $O(nk \log^3 n)$ weak oracle queries. Finally, the time efficiency scales with the total number of weak and strong oracle queries, which leads to the desired lemma statement. □

# 4 Fair Coresets for $(1 + \varepsilon)$-Approximation $k$-median Clustering

In this section, we show how to construct *fair* coresets that preserve the $k$-means and $k$-median costs by $(1 + \varepsilon)$ factor. Limited by the space, we only show the algorithm as in Algorithm 5. We defer the analysis to Appendix C.

---

**Algorithm 5. The construction algorithm for $(1 + \varepsilon)$ coreset**

(1) Run the algorithm of Proposition 2 to get the weak approximate clustering WC.

(2) Initialize all rings $P_{i,j}$ as "*not processed*".

(3) Initialize the fair coreset FC $\leftarrow \emptyset$.

(4) For each center $c_i$ of WC:

  (a) For each ring $P_{i,j}$, let $S_{i,j}$ be some samples from the ring specified later.

  (b) Initialize $\widetilde{P}_i \leftarrow P_i$ as the remaining number of points.

  (c) For $r = 1 : 10 \log^2 n$ iterations:

    i. Sample $s_r = 1000 C_0 \cdot \log^2 n$ points uniformly at random from $\widetilde{P}_i$, and let the sample set be $S_r$.

    ii. Make strong oracle SO queries as follows.
      • For point SO queries, query $\mathsf{SO}(x)$ for all points $x \in S_r$.
      • For distance SO queries, query $\mathsf{SO}(x, c_i)$ for all points $x \in S_r$.

    iii. **Ring assignment:** For each point $x \in S_r$, add $x$ to $S_{i,j}$, where $j$ is the index of the ring to which $x$ belong.

    iv. Find the ring $j^*$ with the *smallest index* such that $|S_{i,j^*}| \geq \frac{4}{5 \log n} s_r$.

    v. Conduct the **sampling and peeling** step as follows.

      A. For each point $x \in \widetilde{P}_i$, run the algorithm of Proposition 3 with the set $S^{\mathsf{peel}} \leftarrow \cup_{j=1}^{j^*} S_{i,j} \cup \{c_i\}$.

      B. Let $T_i^{(r)}$ be the set of points such that $\widetilde{\mathbf{d}}(x, S^{\mathsf{peel}}) \leq (2C_0)^{j^*} \cdot R$.

      C. **Sampling:** sample a set $\mathsf{FC}(T_i^{(r)})$ of $m = \Theta(k\varepsilon^{-2} \log^2 n \log(n\varepsilon^{-1}))$ points from $T_i^{(r)}$ uniformly at random *with replacement*, and weight each sampled points by $\left|T_i^{(r)}\right|/m$.

      D. Add the points in $\mathsf{FC}(T_i^{(r)})$ with the weights to the coreset FC.

      E. **Peeling:** remove all points in $T_i^{(r)}$ from $\widetilde{P}_i$, i.e., $\widetilde{P}_i \leftarrow \widetilde{P}_i \setminus T_i^{(r)}$.

---

Note that in terms of the pseudo-codes, Algorithm 5 is simpler than Algorithm 1 since we do *not* have to handle the charging of rings explicitly with the labels of "*being processed*". We will handle the argument associated with that part in the *analysis*.

Algorithm 5 samples uniformly at random from all the peeled points to form coresets (not only the heavy rings). Intuitively, sampling from the entire set of points is necessary to obtain assignment-preserving coresets. The actual analysis of the algorithm is considerably more involved. The main roadblock here is that almost all existing algorithms for assignment-preserving coresets only work with *rings* (e.g., (Cohen-Addad & Li, 2019; Braverman et al., 2022)), but in our case, we need to work with sets of points that could span multiple rings. This requires us to open the blackbox in (Cohen-Addad & Li, 2019) and re-prove the guarantees for assignment preservation and approximation using the properties of the heavy rings.

In Appendix D, we give a $(1 + \varepsilon)$-coreset for general $(k, z)$-clustering (without fairness constraints). We remark that the algorithmic approach of sampling and peeling from heavy rings with smaller indices first (therefore not ignoring "low-cost" regions similar to Algorithm 5) appears to be applicable to general $(k, z)$-clustering as well. Furthermore, we believe that this could lead to improvements in the dependence on $k$ in the coreset size as well as the number of weak and strong oracle queries. This would require a white-box adaptation of Chen (2009) to work with the non-ring set of points as in line 4(c)vC, and we leave it as an interesting future problem to explore.

# 5 Experiments

We implement and evaluate our algorithm on two real-world datasets for fair $k$-median. We use "Adult" which has about $50,000$ instances and 8 features (Becker & Kohavi, 1996). The second dataset is "Default of Credit Card Clients" with about $30,000$ instances and 9 features (Yeh, 2009). For both we make the sensitive attribute gender.

All experiments were run locally on a Macbook Air. The code can be found here. In both datasets, we first turn each instance into an equivalent numerical representation. Due to computational limits, we first subsampled the dataset to size roughly 2000 using Meyerson sampling (Meyerson, 2001a). Meyerson sampling goes as follows. Start with empty set $S$. Then go through the points in the dataset one by one. Add the first point to $S$. Now for each successive point, add it to $S$ with probability proportional to how far away it is from $S$. Here we define distance as the euclidean distance. If the point is further away, the probability of it being added to $S$ is higher.

We use Meyerson sampling instead of uniform sampling

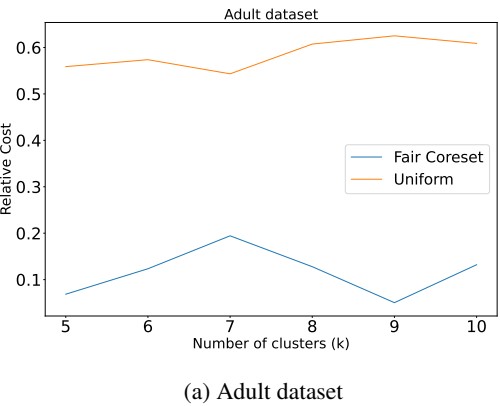
(a) Adult dataset

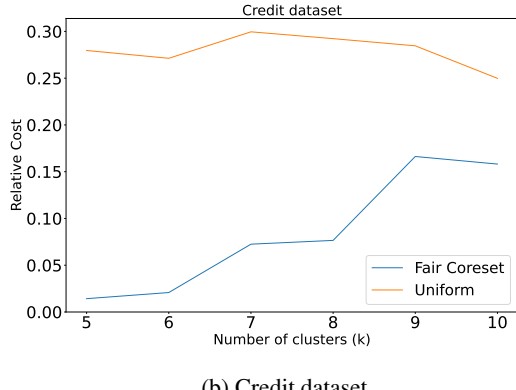
(b) Credit dataset

*Figure 1.* Relative cost of fair $k-$median clustering on real world datasets for different values of $k$

to better preserve the underlying structure of the original dataset. Uniform sampling may entirely miss smaller clusters of points, but Meyerson sampling is more likely to capture them.

After subsampling the original dataset using Meyerson sampling to a smaller dataset, we ran our fair coreset algorithm on this smaller dataset to create a coreset of size $100$. We also made a uniform coreset as a baseline which simply uniformly selected $100$ points.

**Evaluation**    To evaluate the coresets, we use the fairtree algorithm (Backurs et al., 2019), which we run on top of the uniform and our coreset. We also run the fairtree algorithm on the dataset and report the relative cost, i.e. cost reported on dataset minus the cost reported on coreset scaled by the cost on dataset. For all experiments we use the values of balance parameters $p = 1$ and $q = 10$ (refer to Backurs et al. (2019) for more details) and we report the cost averaged over 10 independent runs. As can be seen in Figure 1, our coreset consistently outperforms for all datasets across all values of $k$. Moreover, this is achieved by using only about $200$ SO queries which forms only $10\%$ of the data. We note that for larger datasets we expect the number of SO queries to be a smaller percentage of the size of the dataset.

## Acknowledgements

We thank anonymous ICML reviewers for the insightful comments and suggestions. Samson Zhou was supported in part by NSF CCF-2335411. Hoai-An Nguyen was supported in part by an NSF GRFP fellowship grant number DGE2140739, NSF CAREER Award CCF-2330255, Office of Naval Research award number N000142112647, and a Simons Investigator Award. Prathamesh Dharangutte was supported in part by NSF IIS-2229876, DMS-2220271, DMS-2311064, CCF-2208663, and CCF-2118953.

## Impact Statement

This paper presents work whose goal is to advance the field of Machine Learning. There are many potential societal consequences of our work, none which we feel must be specifically highlighted here.

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

# A    Additional Related Work

We discuss additional related work in the literature, including results on fairness, $k$-clustering and coresets, and weak-strong distance oracles.

**Fairness.**    Chierichetti et al. (2017b) initiated the study of disparate impact in fair clustering, which intuitively seeks each cluster to be balanced in representation across all protected subpopulations. Chierichetti et al. (2017b) initially considered two classes of protected groups, though an active line of research generalized to larger numbers of groups (Rösner & Schmidt, 2018), as well as different clustering objectives (Ahmadian et al., 2019; Bera et al., 2019; Kleindessner et al., 2019b; Schmidt et al., 2019; Cohen-Addad & Li, 2019; Ahmadian et al., 2020a; Esmaeili et al., 2020; 2021; Braverman et al., 2022). A related line of work required that each cluster contained a certain number of representatives from each subpopulation, rather than a certain fraction (Anegg et al., 2022; Jia et al., 2022). More generally, there exist other proposed notions of fairness, including but not limited to (1) social fairness, where the cost function is taken across subpopulations (Jones et al., 2020; Ghadiri et al., 2021; Makarychev & Vakilian, 2021), (2) individual fairness, where seeks to balance the clustering costs of each point in the dataset (Jung et al., 2020; Mahabadi & Vakilian, 2020; Negahbani & Chakrabarty, 2021), and (3) representative fairness, where the number of centers from each class is limited (Kleindessner et al., 2019a; Angelidakis et al., 2022). However, our work specifically considers $(\alpha, \beta)$-fair clustering, i.e., disparate impact, these alternative notions of fairness and others (Ahmadian et al., 2020b; Song et al., 2024; Chen et al., 2025) are not the primary focus of this study.

**$k$-clustering and coresets.**    $k$-clustering is one of the most fundamental problems that has been extensively studied for over six decades (see, e.g., (Forgy, 1965; Lloyd, 1982; Gonzalez, 1985; Charikar et al., 1999; Guha et al., 2003; Kumar et al., 2004; Arthur & Vassilvitskii, 2007; Feldman et al., 2007; Mettu & Plaxton, 2013)). Covering the vast literature for $k$-clustering is impossible for our paper, and we instead focus on the results directly related to ours. For $k$-median and $k$-means, Charikar et al. (1999) gave the first $O(1)$-approximation in polynomial time, and Indyk (1999) gave the first *sublinear* algorithms for $O(1)$-approximation $k$-median, which was later improved by Mettu & Plaxton (2013). There exist certain PTAS algorithms ($(1 + \varepsilon)$-approximation) for $k$-clustering (Kumar et al., 2004; Feldman et al., 2007), although they are exponentially dependent on $k/\varepsilon$. The concept of coresets for $k$-clustering was first introduced by the seminal work of Har-Peled & Mazumdar (2004) for $\mathbb{R}^d$. Later, the influential work by Chen (2009) obtained coresets of size (roughly) $O(k^2 \log^2 n/\varepsilon^2)$ for *general metrics*. The framework of *ring sampling* used by Chen (2009) is quite elegant, and it is the stem of our $(k, \varepsilon)$ coreset for general $(k, z)$ clustering. Later results by Huang & Vishnoi (2020); Cohen-Addad et al. (2021); Braverman et al. (2021; 2022); Huang et al. (2024) further improved the sizes of the coresets on the dependency of $k$, $\log n$, and $\varepsilon$, and the near-optimal coreset sizes have been proven for certain applications (e.g., in Huang et al. (2024)).

**Coresets for fair clustering.**    Since the introduction of fair clustering by Chierichetti et al. (2017a), several works have investigated the construction of coresets that respect fairness constraints. Schmidt et al. (2019) presented a coreset construction for the fair $k$-means problem and extended it to the streaming model. More recently, Bandyapadhyay et al. (2024) proposed coreset constructions for both fair $k$-means and $k$-median clustering using random sampling methods in metric and Euclidean spaces. Their approach yields a constant-factor approximation for metric fair clustering and a $(1 + \varepsilon)$-approximation in Euclidean space, with running time independent of the data dimension.

Further extensions have considered settings where each data point may belong to multiple, possibly overlapping, sensitive groups. For example, a data point representing a person might be labeled with both race and gender, each treated as a sensitive attribute. Huang et al. (2019) gave the first coreset construction for fair $k$-median clustering and also extended it to handle multiple, non-disjoint sensitive attributes. Subsequent works have explored different formulations of fair clustering under overlapping group structures, including Zhang et al. (2024); Gadekar et al. (2025); Thejaswi et al. (2022; 2021; 2024); Chen et al. (2024).

**Weak-strong oracle models.**    In modern ML applications, there have been increasingly common trade-offs between information accuracy and price. For instance, John et al. (2020) studied the bitrate-quality optimization in video transcoding by clustering the videos using meta-information and optimization on a subset of videos with actual clips. Motivated by such applications, there has been a handful of work investigating learning problems with "weak-but-cheap" and "strong-but-expensive" information sources. Apart from the model in Bateni et al. (2024) that studied metric clustering, Silwal et al. (2023) studied *correlation clustering* with the presence of a noisy weak oracle, and Xu et al. (2024) investigated *similarity search* with two types of oracles. Galhotra et al. (2024) proposed a model that is quite close to Bateni et al. (2024), albeit their weak oracle is based on comparing *pairs* of distances. Finally, a handful of work also considered learning with only the

"weak" imprecise oracle (Mazumdar & Saha, 2017; Larsen et al., 2020; Peng & Zhang, 2021), and this setting is closely related to learning-augmented algorithms (see, e.g. Hsu et al. (2019); Indyk et al. (2019); Mitzenmacher & Vassilvitskii (2022); Ergun et al. (2022); Grigorescu et al. (2022); Braverman et al. (2024); C. S. et al. (2024); Braverman et al. (2025); Dong et al. (2024); Fu et al. (2025)).

# B   Deferred Proofs from Section 3

We now give the proof of correctness for Algorithm 1. Our plan is to show that for each iteration of line 4c, $i$). each iteration of line 4(c)vi (and therefore line 4c) only affects the rings with the index at most $j + 1$, and the entire algorithm converges in $O(\log^2 n)$ iterations; $ii$). for any ring $k$ with samples more than $\Omega(k \log^2 n/\varepsilon^2)$, we could get *fairly accurate* weights for the points to form a coreset; and $iii$). for any ring $k$ with samples less than $O(k \log^2 n/\varepsilon^2)$, we can "charge" the cost to the adjacent ring $j$ of this iteration.

**Step I: the convergence of the algorithm and the structural results.** We start with the proof of $i$) which includes the convergence of the algorithm and some structural results that are useful for later steps. We start with a lemma that states the "local" properties of the peeling step.

**Lemma B.1.** *Let $j^*$ be the index of the ring found by line 4(c)iv of Algorithm 1. Then, with probability at least $1 - 1/\operatorname{poly}(n)$, no points in $P_{i,\ell}$ for $\ell > j^* + 1$ is removed in the peeling step.*

*Proof.* The lemma follows from the guarantees in Proposition 3 and the choice of the multiplicative $C_0$ factor in the way we partition the rings. Concretely, let us analyze any fixed index $\ell > j^* + 1$. Note that for the ball formed by points in $S^{\mathsf{peel}}$, there are at least $80 \cdot C_0 \cdot \frac{k \log^2 n}{\varepsilon^3} \geq 10 \log n$ sampled points. Furthermore, by the definition of rings outside $j^* + 1$, we have $\mathbf{d}(x, c_i) > (2C_0)^{j^*+1} \cdot R$. Therefore, for all points $x \in P_{i,\ell}$, we have that

$$\widetilde{\mathbf{d}}(x, S^{\mathsf{peel}}) \geq \mathbf{d}(x, c_i) > (2C_0)^{j^*+1} \cdot R$$

with probability at least $1 - 1/\operatorname{poly}(n)$ by Proposition 3. Therefore, by the peeling rule of our algorithm, no such point $x$ could be removed. $\qquad\square$

Lemma B.1 ensures that the peeling process never affects the points outside the ring with index $j^* + 1$. For the purpose of our analysis, we will need to handle ring $j = 0$ with more care. Next, we show a lemma that characterizes the behavior of the peeling for rings with $j = 0$.

**Lemma B.2.** *With probability at least $1 - 1/\operatorname{poly}(n)$, Algorithm 4 (the CONSERVATIVE-PEELING algorithm) could only be executed once. Furthermore, after the execution of Algorithm 4, we have that*

- *all points $x$ such that $\mathbf{d}(x, c_i) \leq \frac{R}{2C_0}$ are removed from $\widetilde{P}_i$.*

- *no points in $P_{i,1}$ is removed.*

*Proof.* We let $P_{i,0}^{\mathsf{close}}$ be the set of points such that for $x \in S_{i,0}^{\mathsf{close}}$, $\mathbf{d}(x, c_i) \leq \frac{R}{2C_0}$, and let $S_{i,0}^{\mathsf{close}}$ be the sampled set from $P_{i,0}^{\mathsf{close}}$. Note that for Algorithm 4 to be executed, there must be

$$\left|S_{i,0}^{\mathsf{close}}\right| \geq \frac{1}{2} \cdot |S_{i,0}| \geq 40 \cdot C_0 \cdot \frac{k \log^2 n}{\varepsilon^3} \geq 10 \log n,$$

so that it is possible to find a set of $10 \log n$ points. Therefore, when running the algorithm of Proposition 3, we have $R_S \leq R/2C_0$. Therefore, conditioning on the high probability of Proposition 3, every point $x \in P_{i,0}^{\mathsf{close}}$, we have that

$$\widetilde{\mathbf{d}}(x, S^{\mathsf{peel}}) \leq \mathbf{d}(x, c_i) + C_0 \cdot R_S \leq R.$$

Therefore, all points in $P_{i,0}^{\mathsf{close}}$ is removed. Furthermore, for each point $y \in P_{i,1}$, we have that $\widetilde{\mathbf{d}}(y, S^{\mathsf{peel}}) \geq \mathbf{d}(y, c_i) \geq R$, which means such points will *not* be peeled. Finally, since all points in $P_{i,0}^{\mathsf{close}}$ are peeled after one execution of Algorithm 4, the condition of invoking Algorithm 4 will never be met, which means the algorithm is executed at most once. $\qquad\square$

We further provide another lemma showing that we could *remove* all points *within* the radius of ring $j^*$. This lemma is crucial for the *fast convergence* of the algorithm.

**Lemma B.3.** *Let $j^*$ be the index of the ring found by line 4(c)iv of Algorithm 1. Then, with probability at least $1-1/\operatorname{poly}(n)$, all points in $P_{i,\ell}$ for $\ell \leq j^*$ are removed in the peeling step.*

*Proof.* Similar to Lemma B.1, Lemma B.3 is an application of Proposition 3. For any ring with level $\ell \leq j^*$. Let $q$ be the smallest ring index such that the ring has *not* been marked as "*processed*". For the ball formed by points in $\cup_{z=q}^{j^*} P_q$, there are at least $80 \cdot C_0 \cdot \frac{k \log^2 n}{\varepsilon^3} \geq 10 \log n$ sampled points, and radius of the sampled points is at most $(2C_0)^{j^*} \cdot R$. As such, by Proposition 3, for any point $s \in \cup_{z=q}^{j^*} P_q$, with probability at least $1 - 1/\operatorname{poly}(n)$, we have that

$$\widetilde{\mathbf{d}}(x, S^{\mathsf{peel}}) \leq \mathbf{d}(x,s) + C_0 \cdot (2C_0)^{j^*} \cdot R = \left((2C_0)^{j^*} + C_0 \cdot (2C_0)^{j^*}\right) \cdot R \leq (2C_0)^{j^*+1} \cdot R.$$

Therefore, by the rules of the algorithm, all points in rings $P_{i,\ell}$ such that $\ell \leq j^*$ will be removed by the peeling step with probability at least $1 - 1/\operatorname{poly}(n)$. $\qquad\square$

There are several consequences of Lemmas B.1 to B.3: by Lemma B.1 and the algorithm design, each ring $P_{i,j}$ is added to coreset by *exactly one* iteration; furthermore, by Lemma B.2 and Lemma B.3, our algorithm could *converge* in $O(\log^2 n)$ iterations. We formalize this statement as the following lemma.

**Lemma B.4.** *With probability at least $1 - 1/\operatorname{poly}(n)$, the following statements for Algorithm 1 are true:*

- *Each ring $P_{i,j}$ is marked as "processed" by exactly one iteration of line 4c.*

- *After $O(\log^2 n)$ iterations of line 4c, all points in $P_i$ is marked as "processed".*

*Proof.* We first observe that for a fixed iteration $r$, once the algorithm has identified $j^*$, there are two cases

(1) either $j^*$ has been marked "*processed*", for which case we should have $j^* = j' + 1$, where $j'$ is the picked ring index of the last iteration $r - 1$ by line 4(c)iv (i.e, $j'$ is the "$j^*$" of the last iteration);

(2) or $j^*$ has is marked "*not processed*", which means $j^* > j' + 1$.

To elaborate the relationship between the cases and the value of $j^*$, note that conditioning on the success of Lemma B.2 and Lemma B.3, no points in the rings $P_{i,\ell}$ for $\ell < j'$ will survive in $\widetilde{P}_i$ after round $r - 1$. Furthermore, by the design of the algorithm, no rings with index $\ell > j' + 1$ will be marked as "*processed*" before round $r$.

We now prove the second bullet first since it will be the basis for a union-bound argument for the first bullet. We claim that ring $j^*$ accounts for at least $\Omega(1/\log n)$ fraction of the points in $\widetilde{P}_i$, formalized as follows

**Claim B.5.** *For any iteration $r$, let $j^*$ be the ring found by line 4(c)iv. Then, with probability at least $1 - 1/\operatorname{poly}(n)$, we have that*

$$\left|P_{i,j^*} \cap \widetilde{P}_i\right| \geq \frac{1}{50 \log n} \cdot \left|\widetilde{P}_i\right|.$$

*Proof.* The claim is a simple application of Chernoff bound. Let $I_x$ be the random variable for a point $x$ to be sampled, and the random variable $I_{j^*} := \sum_{x \in P_{i,j^*} \cap \widetilde{P}_i} I_x$ is the number of total points we samples from ring $j^*$ for the current iteration. Since we sample uniformly at random, we have that

$$\mathbb{E}\left[I_{j^*}\right] = \frac{\left|P_{i,j^*} \cap \widetilde{P}_i\right|}{\left|\widetilde{P}_i\right|} \cdot 100 C_0 \cdot \frac{k \log^3 n}{\varepsilon^3}.$$

Therefore, if $\left|P_{i,j^*} \cap \widetilde{P}_i\right| < \frac{1}{50\log n} \cdot \left|\widetilde{P}_i\right|$, we will have $\mathbb{E}\left[I_{j^*}\right] \leq 2C_0 \cdot \frac{k\log^2 n}{\varepsilon^3}$. On the other hand, by the design of line 4(c)iv, the ring $j^*$ should have at least $80C_0 \cdot \frac{k\log^2 n}{\varepsilon^3}$ points. By an application of Chernoff bound, we have that

$$
\Pr\left(I_{j^*} \geq 80C_0 \cdot \frac{k\log^2 n}{\varepsilon^3} \,\middle|\, \left|P_{i,j^*} \cap \widetilde{P}_i\right| < \frac{1}{50\log n} \cdot \left|\widetilde{P}_i\right|\right)
$$

$$
\leq \Pr\left(I_{j^*} \geq (1+\delta) \cdot \mathbb{E}\left[I_{j^*}\right] \,\middle|\, \delta \cdot \mathbb{E}\left[I_{j^*}\right] \geq 78 \cdot C_0 \cdot \frac{k\log^2 n}{\varepsilon^3},\ \mathbb{E}\left[I_{j^*}\right] \leq 2C_0 \cdot \frac{k\log^2 n}{\varepsilon^3}\right)
$$

$$
\leq 1/\operatorname{poly}(n).
$$

As such, conditioning on the above high-probability event, we have that for $j^*$ to have at least $80C_0 \cdot \frac{k\log^2 n}{\varepsilon^3}$ points, there must be $\left|P_{i,j^*} \cap \widetilde{P}_i\right| \geq \frac{1}{50\log n} \cdot \left|\widetilde{P}_i\right|$, which is as claimed. $\hfill$ Claim B.5 $\square$

Note that by a simple averaging argument, since there are at most $\log n$ rings for each center, by an averaging argument, there exists at least one ring with at least $80C_0 \cdot \frac{k\log^2 n}{\varepsilon^3}$ samples. Hence, in both cases for $j^* = j' + 1$ and $j^* > j' > 1$, as long as the algorithm does *not* invoke Algorithm 4, *peeling* step, at least all the points in $P_{i,j^*} \cap \widetilde{P}_i$ are removed. As such, we have that at step $r$, with probability at least $1 - \operatorname{poly}(n)$, there is

$$
\left|\widetilde{P}_i^{(r+1)}\right| \leq \left(1 - \frac{1}{50\log n}\right) \cdot \left|\widetilde{P}_i^{(r)}\right|,
$$

where notation $\widetilde{P}_i^{(r)}$ and $\widetilde{P}_i^{(r+1)}$ are used to denote the surviving points ($\widetilde{P}_i$) for rounds $r$ and $(r+1)$. Furthermore, by Lemma B.2, Algorithm 4 could be invoked for at most once. Therefore, after $O(\log^2 n)$ rounds, and conditioning on the high-probability events to happen over *all* the rounds (which happens with probability at least $1 - 1/\operatorname{poly}(n)$ by a union bound), all points are removed from $\widetilde{P}_i$ and marked as "*processed*".

We now turn to the proof of the first bullet. Note that in both cases of $j^* = j' + 1$ and $j^* > j' + 1$ for a fixed iteration $r$, the rings that are marked as "*processed*" are $[j' + 2, j^* + 1]$ (if $j^* = 0$ and Algorithm 4 is executed, then only $j^*$ is marked as "*processed*"). Therefore, by a union bound on the high probability events, with probability at least $1 - 1/\operatorname{poly}(n)$, each ring has a unique iteration that marked it as "*processed*". $\hfill$ Lemma B.4 $\square$

**Step II: handling the rings with many samples.** We now handle the rings that are marked "*processed*" by Algorithm 2 (line 4(c)v). The analysis is similar to Chen (2009), except we now need to deal with an approximate number of points for weighing the coreset points. First, we show that we can approximate the number of points within these rings with small relative error.

**Lemma B.6.** *Let $S$ with $s_r = O(k\log^3 n/\varepsilon^3)$ be the set of points sampled uniformly at random from $\widetilde{P}_i$ and let $S_{i,j} = P_{i,j} \cap S$. For all $j$ such that $|S_{i,j}| \geq 30k\log^2 n/\varepsilon^2$, let $S_{i,j}^{\mathrm{est}}$ be any 1/3 fraction of points from $S_{i,j}$, and $\widetilde{m}_{i,j} = \frac{3|\widetilde{P}_i|}{s_r} \cdot \left|S_{i,j}^{\mathrm{est}}\right|$. For all $|P_{i,j}| \geq 30\log n/\varepsilon^2$, with probability at least $1 - \frac{1}{\operatorname{poly} n}$ we have $\widetilde{m}_{i,j} \in [(1-\varepsilon)|P_{i,j}|, (1+\varepsilon)|P_{i,j}|]$.*

*Proof.* By the design of Algorithm 1, Algorithm 2 is only run on $S_{i,j}$ that have at least $30k\log^2 n/\varepsilon^2$ points in them. Since we run SO queries on $S$, we know the exact set $P_{i,j} \cap S$. For $\ell \in [|P_{i,j}|]$, let $X_\ell$ be a random variable that has value 1 if point $\ell$ of $|P_{i,j}|$ was sampled and added to $S$ and let $X = \sum_\ell X_\ell$. We have

$$
\mathbb{E}\left[\left|S_{i,\ell}^{\mathrm{est}}\right|\right] = \frac{\mathbb{E}[X]}{3} = \frac{s_r}{3} \cdot \frac{|P_{i,j}|}{|\widetilde{P}_i|}.
$$

By the definition of $\widetilde{m}_{i,j}$, we have $\mathbb{E}[\widetilde{m}_{i,j}] = |P_{i,j}|$. Now, applying Chernoff bound we get

$$
\Pr\left(|\widetilde{m}_{i,j} - |P_{i,j}|| \geq \varepsilon \cdot |P_{i,j}|\right) \leq 2\exp\left(-\frac{\varepsilon^2|P_{i,j}|}{3}\right) \leq 2\exp\left(-\frac{30\log n}{3}\right) \leq \frac{1}{\operatorname{poly} n}.
$$

Finally, a union bound over at most $O(k\operatorname{polylog} n)$ such $P_{i,j}$ completes the proof. $\hfill \square$

For coreset guarantees, we need to bound the difference in cost for points in ring and the coreset points due to arbitrary set of centers. The proof is an extension of result from Chen (2009), which itself relies on the following result from Haussler (1992).

**Lemma B.7.** *Let $M \geq 0$ and $\eta$ be fixed constants, and let $h(.)$ be a function defined on set $V$ such that $\eta \leq h(p) \leq \eta + M$ for all $p \in V$. Let $U \{p_1, \cdots, p_s\}$ be a set of $s$ samples drawn independently and uniformly form $V$, and let $\delta > 0$ be a parameter. If $s \geq (M^2/2\delta^2) \ln(2/\lambda)$, then*

$$\Pr\left[\left|\frac{h(V)}{|V|} - \frac{h(U)}{|U|}\right| \geq \delta\right] \leq \lambda,$$

*where $h(U) = \sum_{u \in U} h(u)$ and $h(V) = \sum_{v \in V} h(v)$.*

We now prove the approximation guarantee of the coreset points that are added by Algorithm 2.

**Lemma B.8.** *For a ring $P_{i,\ell}$ that is marked "processed" by Algorithm 2, let $\mathcal{C}$ be any set of centers. Let $S_{i,\ell}$ denote the set of points in $S \cap P_{i,\ell}$ that are uniformly sampled and $S_{i,\ell}^{\mathsf{weight}}$ be any 2/3 points in $S_{i,\ell}$ that are assigned weight $w = \widetilde{m}_{i,\ell}/|S_{i,\ell}^{\mathsf{weight}}|$ and added to $\mathsf{SC}$. Then, with probability at least $1 - \lambda$*

$$\left|\mathrm{cost}(\mathcal{C}, P_{i,\ell}) - \mathrm{cost}(\mathcal{C}, S_{i,\ell}^{\mathsf{weight}})\right| \leq 2\varepsilon |P_{i,\ell}| \mathrm{diam}(P_{i,\ell})$$

*Proof.* For a point $p \in P_{i,\ell}$, let $h(p) = \mathbf{d}(\mathcal{C}, p)$. We have,

$$\mathbf{d}(\mathcal{C}, P_{i,\ell}) \leq h(p) = \mathbf{d}(\mathcal{C}, p) \leq \mathbf{d}(\mathcal{C}, P_{i,\ell}) + \mathrm{diam}(P_{i,\ell}).$$

Let $\eta = \mathbf{d}(\mathcal{C}, P_{i,\ell})$, $M = \mathrm{diam}(P_{i,\ell})$ and $\delta = \varepsilon M$. For $\left|S_{i,\ell}^{\mathsf{weight}}\right| \geq (M^2/2\delta^2) \ln 2/\lambda$, from Lemma B.7 we have

$$\Pr\left[\left|\frac{\sum_{p \in P_{i,\ell}} \mathbf{d}(\mathcal{C}, p)}{|P_{i,\ell}|} - \frac{\sum_{s \in S_{i,\ell}^{\mathsf{weight}}} \mathbf{d}(\mathcal{C}, s)}{\left|S_{i,\ell}^{\mathsf{weight}}\right|}\right| \geq \varepsilon \cdot \mathrm{diam}(P_{i,\ell})\right] \leq \lambda \tag{1}$$

Using above equation and conditioning on the high probability event of Lemma B.6, we get

$$\begin{aligned}
\left|\mathrm{cost}(\mathcal{C}, P_{i,\ell}) - \mathrm{cost}(\mathcal{C}, S_{i,\ell}^{\mathsf{weight}})\right| &= \left|\sum_{p \in P_{i,\ell}} \mathbf{d}(\mathcal{C}, p) - \sum_{s \in S_{i,\ell}^{\mathsf{weight}}} \mathbf{d}(\mathcal{C}, s) w(s)\right| \\
&= |P_{i,\ell}| \left|\frac{\sum_{p \in P_{i,\ell}} \mathbf{d}(\mathcal{C}, p)}{|P_{i,\ell}|} - \frac{\sum_{s \in S_{i,\ell}^{\mathsf{weight}}} \mathbf{d}(\mathcal{C}, s) w(s)}{|P_{i,\ell}|}\right| \\
&\leq |P_{i,\ell}| \left|\frac{\sum_{p \in P_{i,\ell}} \mathbf{d}(\mathcal{C}, p)}{|P_{i,\ell}|} - \frac{(1-\varepsilon) \sum_{s \in S_{i,\ell}^{\mathsf{weight}}} \mathbf{d}(\mathcal{C}, s) |P_{i,\ell}|}{|P_{i,\ell}| \left|S_{i,\ell}^{\mathsf{weight}}\right|}\right| \\
&\qquad\qquad\qquad\qquad \text{(From Lemma B.6 and } w = \widetilde{m}_{i,\ell}/|S_{i,\ell}^{\mathsf{weight}}|) \\
&\leq |P_{i,\ell}| \left|\frac{\sum_{p \in P_{i,\ell}} \mathbf{d}(\mathcal{C}, p)}{|P_{i,\ell}|} - \frac{\sum_{s \in S_{i,\ell}^{\mathsf{weight}}} \mathbf{d}(\mathcal{C}, s)}{\left|S_{i,\ell}^{\mathsf{weight}}\right|}\right| + \varepsilon |P_{i,\ell}| \frac{\sum_{s \in S_{i,\ell}^{\mathsf{weight}}} \mathbf{d}(\mathcal{C}, s)}{\left|S_{i,\ell}^{\mathsf{weight}}\right|} \\
&\leq |P_{i,\ell}| \varepsilon \mathrm{diam}(P_{i,\ell}) + \varepsilon |P_{i,\ell}| \mathrm{diam}(P_{i,\ell}) \\
&\leq 2\varepsilon |P_{i,\ell}| \mathrm{diam}(P_{i,\ell})
\end{aligned}$$

w.p. at least $1 - \lambda$. Second last inequality follows from, for $s \in S_{i,\ell}^{\mathsf{weight}}$, $\mathbf{d}(\mathcal{C}, s) \leq \mathrm{diam}(P_{i,\ell})$. □

The above bounds the cost for rings that have sufficiently many points sampled in them ($\geq M^2/2\delta^2 \ln(2/\lambda)$). We now show that the number of points added by Algorithm 1 are enough for us to use the above lemma (for appropriately chosen $\lambda$), and prove our main lemma for the rings with points added to the strong coreset SC.

**Lemma B.9.** *Let $\mathcal{P}^{\text{heavy}}$ be the set of rings with points being added to SC. Furthermore, let*

$$\text{cost}(\mathcal{C}, \mathcal{P}^{\text{heavy}}) := \sum_{P_{i,j} \in \mathcal{P}^{\text{heavy}}} \text{cost}(\mathcal{C}, P_{i,j})$$

*be the total cost with respect to any set of center $\mathcal{C} \subseteq X$ of size at most $k$, induced by the rings in $\mathcal{P}^{\text{heavy}}$, and let*

$$\widetilde{\text{cost}}(\mathcal{C}, \mathcal{P}^{\text{heavy}}) := \sum_{P_{i,j} \in \mathcal{P}^{\text{heavy}}} \text{cost}(\mathcal{C}, S_{i,j}^{\text{weight}}).$$

*Then, with probability at least $1 - 1/\text{poly}(n)$, we have*

$$\left| \text{cost}(\mathcal{C}, \mathcal{P}^{\text{heavy}}) - \widetilde{\text{cost}}(\mathcal{C}, \mathcal{P}^{\text{heavy}}) \right| \leq 2\varepsilon \cdot \sum_{P_{i,j} \in \mathcal{P}^{\text{heavy}}} |P_{i,j}| \operatorname{diam}(P_{i,j}).$$

*Proof.* Fix some set of centers $\mathcal{C}$ of size at most $k$. Let $\lambda = \Lambda / \left( n^k (ck \log \beta n) \right)$ for some constant $c$. From Lemma B.8 for a particular $i$ and $\ell$, we have with probability at least $1 - \lambda$

$$\left| \text{cost}(\mathcal{C}, P_{i,\ell}) - \text{cost}(\mathcal{C}, S_{i,\ell}^{\text{weight}}) \right| \leq 2\varepsilon |P_{i,\ell}| \operatorname{diam}(P_{i,\ell})$$

This holds for the points added to SC by Algorithm 2 as the number of points added for each heavy ring is $\left| S_{i,\ell}^{\text{weight}} \right| \geq 30 \cdot k \log^2 n / \varepsilon^2 \geq (1/2\varepsilon^2) \ln 2/\lambda = (1/2\varepsilon^2) \left( \ln(cn^k k \log \beta n) + \ln \frac{1}{\Lambda} \right)$

Now, since we have $k$ centers in the approximate solution and each center has at most $O(\log \beta n)$ rings, union bound over all the rings gives us with probability at least $1 - \Lambda/n^k$,

$$\left| \text{cost}(\mathcal{C}, \mathcal{P}^{\text{heavy}}) - \widetilde{\text{cost}}(\mathcal{C}, \mathcal{P}^{\text{heavy}}) \right| \leq \sum_{P_{i,\ell} \in \mathcal{P}^{\text{heavy}}} \left| \text{cost}(\mathcal{C}, P_{i,\ell}) - \text{cost}(\mathcal{C}, S_{i,\ell}^{\text{weight}}) \right|$$

$$\leq 2\varepsilon \cdot \sum_{P_{i,\ell} \in \mathcal{P}^{\text{heavy}}} |P_{i,\ell}| \operatorname{diam}(P_{i,\ell})$$

As there are at most $n^k$ choices of $k$ centers $\mathcal{C}$ from point set $X$ of size $n$, the proof concludes with a final union bound over the choice of centers $\mathcal{C}$ and setting $\Lambda = 1/n^{c'}$. □

**Step III: handling the rings with few samples.** We now turn to the analysis of the rings that are "neglected" by line 4(c)v of Algorithm 1. We do *not* have any subroutine that handles the cost of the rings with samples less than $30 \cdot \frac{k \log^2 n}{\varepsilon^2}$. Our plan for these rings is to show that the number of points there must be small, and as such, the contributions to the total cost must be small. Therefore, we can charge the cost of these rings to the cost of the rings with enough samples.

To that end, we show the following lemma regarding the *actual size* between the rings with a few vs. many points.

**Lemma B.10.** *Let ring $P_{i,\ell}$ be marked as "processed" in iteration $r$ without any point $x \in P_{i,\ell}$ being added to the coreset SC. Furthermore, let $j^*$ be the ring found by line 4(c)iv of iteration $r$. Then, with probability at least $1 - 1/\text{poly}(n)$, we have*

$$|P_{i,j^*}| \geq \frac{2C_0}{\varepsilon} \cdot |P_{i,\ell}|.$$

*Proof.* Let us say that ring $P_{i,j^*}$ has $y_1$ points in $\tilde{P}_i$. Therefore, we have that $\mathbb{E}[|S_{i,j^*}|] = y_1 \cdot \frac{s_r}{\tilde{P}_i}$. Therefore by a Chernoff bound we have that $|S_{i,j^*}| \leq (1+\varepsilon)y_1 \cdot \frac{s_r}{\tilde{P}_i}$ with probability at least $1 - 1/\text{poly}(n)$. This gives us that with high probability

$y_1 \geq \frac{|S_{i,j^*}|}{(1+\varepsilon)} \cdot \frac{\tilde{P}_i}{s_r}$. Similarly, let us say that ring $P_{i,j}$ has $y_2$ points in $\tilde{P}_i$. Therefore, we have that $\mathbb{E}\left[|S_{i,j}|\right] = y_2 \cdot \frac{s_r}{\tilde{P}_i}$. Therefore by a Chernoff bound we have that $y_2 \leq \frac{|S_{i,j}|}{(1-\varepsilon)} \cdot \frac{s_r}{\tilde{P}_i}$.

We now from the algorithm that $|S_{i,j^*}| \geq 80C_0 \cdot \frac{k \log^2 n}{\varepsilon^3}$ and $|S_{i,j}| \leq 30 \cdot \frac{k \log^2 n}{\varepsilon^2}$. Therefore we have the result. $\qquad \square$

For each ring $P_{i,\ell}$ marked as "*processed*" without points being added to $\mathsf{SC}$, we define $j^*(\ell)$ as be the index of ring $P_{i,j^*}$ that marked $P_{i,\ell}$ as "*processed*".

We now define the following *charging scheme* for the quantity $|P_{i,\ell}| \cdot \mathrm{diam}(P_{i,\ell})$, which will distribute the cost to $|P_{i,\ell}| \cdot \mathrm{diam}(P_{i,\ell})$.

---

**Algorithm 6.  A charging scheme** (a thought process for the **analysis purpose** only).

- For each ring $P_{i,\ell}$ such that $\ell$ is marked as "*processed*" in iteration $r$ without any point $x \in P_{i,\ell}$ being added to the coreset $\mathsf{SC}$:

    ○ If $j^*(\ell) \neq 0$, we write a charge of $\frac{\varepsilon}{2C_0} \cdot \mathrm{diam}(P_{i,\ell})$ to *all* points in $P_{i,j^*(\ell)}$.
    ○ Otherwise, if $j^*(\ell) = 0$ (which implies $\ell = 1$), we write a charge of $\frac{\varepsilon}{2C_0} \cdot \mathrm{diam}(P_{i,1})$ to *all* points in $P_{i,0} \setminus P_{i,0}^{\mathrm{close}} := \{x \in P_{i,0} \mid \mathbf{d}(x, c_i) \geq \frac{R}{2C_0}\}$.

- If ring $P_{i,\ell}$ receives charges of at most $\gamma$ on each point *and* $\ell$ is marked as "*processed*" without any point $x \in P_{i,\ell}$ being added to the coreset $\mathsf{SC}$, then *transfer* the charges by writing charges of $\frac{\varepsilon}{2C_0} \cdot \gamma$ to *all* points in $P_{i,j^*(\ell)}$.

- Continue recursively until all charges are written on points of rings $P_{i,\ell}$ such that points $x \in P_{i,\ell}$ are added to the coreset $\mathsf{SC}$.

---

Note that Algorithm 6 is a *thought process*: we cannot hope to actually perform the operations without making many strong oracle queries. However, such a thought process as an analytical tool is valid. Also note that Algorithm 6 eventually terminates since there has to be at least a ring $j^*$ (of the first iteration) that has points added to the coreset $\mathsf{SC}$. We now give two lemmas that characterize the guarantees of the charging scheme.

**Lemma B.11.** *For any ring $P_{i,\ell}$, conditioning on the high-probability event of Lemma B.10, the total number of charges Algorithm 6 could distribute is at least $|P_{i,\ell}| \cdot \mathrm{diam}(P_{i,\ell})$.*

*Proof.* The lemma is a direct consequence of Lemma B.10. For each ring $P_{i,\ell}$, we claim that conditioning on the high-probability event of Lemma B.10, there is $\left|P_{i,j^*(\ell)}\right| \geq \frac{C_0}{50\varepsilon} \cdot |P_{i,\ell}|$. The statement trivially holds if $j^*(\ell) \neq 0$. On the other hand, if $j^*(\ell) = 0$, by the rules in Algorithm 1 (which implies $\ell = 1$), we argue that there are $\frac{C_0}{50\varepsilon} \cdot |P_{i,1}|$. In particular, we could treat $P_{i,0} \setminus P_{i,0}^{\mathrm{close}}$ (defined in Algorithm 6) as a separate ring, and apply Lemma B.10 to get the desired result. Therefore, the total charge $P_{i,\ell}$ could write to is at least

$$\frac{\varepsilon}{2C_0} \cdot \mathrm{diam}(P_{i,\ell}) \cdot \left|P_{i,j^*(\ell)}\right| \geq \mathrm{diam}(P_{i,\ell}) \cdot |P_{i,\ell}|,$$

as claimed. $\qquad \square$

Next, we bound the number of charges for any point in a ring that could possibly be received.

**Lemma B.12.** *For any ring $P_{i,\ell}$, conditioning on the high-probability event of Lemma B.10, for any point $x \in P_{i,\ell}$, the total number of charges written on $x$ by Algorithm 6 is at most $2\varepsilon \cdot \mathrm{diam}(P_{i,\ell+1})$ for any $\varepsilon < 1$.*

*Proof.* For simplicity let us assume for now that no *recursive charging* happens. Note that for each ring $j^*$, only rings with indices $q \in [0, j^* + 1]$ could possibly write charges to it. Let $\mathsf{charge}(j^* \leftarrow q)$ be the number of charges each point in ring $j^*$ could receive from ring $q$. For ring $j^* + 1$, we have that

$$\mathsf{charge}(j^* \leftarrow j^* + 1) \leq \frac{\varepsilon}{2C_0} \cdot \mathrm{diam}(P_{i,j^*+1})$$

by the rules of charging. Furthermore, for each ring $q \in [o, j^* - 1]$, we have that

$$\mathsf{charge}(j^* \leftarrow q) \leq \frac{\varepsilon}{2C_0} \cdot \mathrm{diam}(P_{i,q}) \leq \frac{\varepsilon}{2C_0} \cdot \mathrm{diam}(P_{i,j^*+1}) \cdot (\frac{1}{2C_0})^{j^*-q+1},$$

where the last inequality follows from the definition of the rings. As such, the total charges on ring $j^*$ could be bounded by

$$\sum_q \mathsf{charge}(j^* \leftarrow q) \leq \sum_{q=0}^{j^*+1} \frac{\varepsilon}{2C_0} \cdot \mathrm{diam}(P_{i,q})$$

$$\leq \frac{\varepsilon}{2C_0} \cdot \mathrm{diam}(P_{i,j^*+1}) \cdot \sum_{q=0}^{j^*+1} \cdot (\frac{1}{2C_0})^{j^*-q+1}$$

$$\leq \frac{\varepsilon}{C_0} \cdot \mathrm{diam}(P_{i,j^*+1}), \qquad (\textstyle\sum_{q=0}^{j^*+1} \cdot (\frac{1}{2C_0})^{j^*-q+1} \leq 2 \text{ for any } j^* \geq 0)$$

which gives us the desired results for non-recursive charges.

We now handle recursive charges. By our charging scheme in Algorithm 6 and Lemma B.4, only adjacent rings transfer charges. To elaborate, the only case that a ring could become $j^*$ of that iteration without having added points to SC is that it was marked as "*processed*" by the $j^*$ of the last iteration (the $j^* = j' + 1$ case in the proof of Lemma B.4). Therefore, by our analysis of the non-recursive sharing, the amount of charges ring $\ell$ could transfer to ring $\ell - 1$ for one level of recursion are at most

$$\frac{\varepsilon}{2C_0} \cdot \frac{\varepsilon}{C_0} \cdot \mathrm{diam}(P_{i,\ell+1}) \leq \varepsilon^2 \cdot \frac{1}{2C_0^2} \cdot \mathrm{diam}(P_{i,\ell}) \cdot 2C_0 \leq \varepsilon^2 \cdot \mathrm{diam}(P_{i,\ell}).$$

Hence, the total amount of charges that could be transferred to any such $j^*$ is at most

$$\sum_{\ell=j^*+1}^{\log n} \varepsilon^{2 \cdot (j^*-\ell)} \cdot \mathrm{diam}(P_{i,j^*}) \leq \varepsilon \cdot \mathrm{diam}(P_{i,j^*+1}).$$

Therefore, the charge a layer $j^*$ could receive due to the transfer from other layers is at most $\varepsilon \cdot \mathrm{diam}(P_{i,j^*+1})$. Combining this with the charging upper-bound of the non-recursive case gives us the desired statement. $\qquad\square$

We now use Lemma B.11 and Lemma B.12 to bound the total cost induced by the rings that are marked as "*processed*" without points added to the coreset SC.

**Lemma B.13.** *Let $P_{i,\ell}$ be a ring such without points added to SC, and $P_{i,j^*(\ell)}$ be the ring for $P_{i,\ell}$ to charge to in the charging scheme of Algorithm 6. Then, we have that*

$$|P_{i,\ell}| \cdot \mathrm{diam}(P_{i,\ell}) \leq \varepsilon \cdot C_0 \cdot |P_{i,j^*(\ell)}| \cdot \mathrm{diam}(P_{i,j^*(\ell)}).$$

*Proof.* By Lemma B.11, each of such ring $P_{i,\ell}$ could distribute all the costs as the charges. By Lemma B.12, any point in such ring $P_{i,j^*(\ell)}$ receives at most $2\varepsilon \cdot \mathrm{diam}(P_{i,j^*(\ell)+1})$ charges. We now claim that $\mathrm{diam}(P_{i,j^*(\ell)+1}) \leq 2C_0^2 \cdot \mathrm{diam}(P_{i,j^*(\ell)})$. For all rings *except* $j^*(\ell) = 0$, the statement simply follows from the construction. If $j^*(\ell) = 0$, by the rule of Algorithm 1, the charges could only be written on $P_{i,0} \setminus P_{i,0}^{\mathrm{close}}$ by the charging rule. Therefore, we have $\mathrm{diam}(P_{i,j^*(\ell)+1}) \leq 2C_0^2 \cdot \mathrm{diam}(P_{i,j^*(\ell)})$, and a summation over the charges on the points gives the desired lemma statement. $\qquad\square$

We now use Lemma B.13 to bound the cost of the coreset. We first present the following lemma that uses the quantity $|P_{i,j}| \cdot \mathrm{diam}(P_{i,j})$ to bridge the cost between the rings with points in SC and the rings without.

**Lemma B.14.** *Let $\mathcal{C}$ be any set of centers. Then, for any ring $P_{i,j^*}$ and rings $P_{i,j}$ such that $j \leq j^* + 1$ and $|P_{i,j}| \leq \eta \cdot |P_{i,j^*}|$, we have that*

$$\mathrm{cost}(\mathcal{C}, P_{i,j}) \leq \eta \cdot \mathrm{cost}(\mathcal{C}, P_{i,j^*}) + 2C_0 \cdot |P_{i,j}| \cdot \mathrm{diam}(P_{i,j}).$$

*Proof.* The lemma stems from the triangle inequality. Concretely, let $x \in P_{i,j}$ be a point in the ring $P_{i,j}$, we could find a corresponding a point $y \in P_{i,j^*}$ assigned to center $c \in \mathcal{C}$ such that

$$
\begin{aligned}
\text{cost}(\mathcal{C}, x) &\leq \mathbf{d}(c, x) && (\text{cost}(\mathcal{C}, x) \text{ is the optimal cost for } x) \\
&\leq \mathbf{d}(c, y) + \mathbf{d}(x, y) && (\text{triangle inequality}) \\
&\leq \text{cost}(\mathcal{C}, y) + \text{diam}(P_{i,j^*}) \\
&\leq \text{cost}(\mathcal{C}, y) + 2C_0 \cdot \text{diam}(P_{i,j}). && (\text{by our construction of the rings and } j \leq j^* + 1)
\end{aligned}
$$

Therefore, we could "reuse" the least-cost point $y \in P_{i,j^*}$ for $|P_{i,j}| \leq \eta \cdot |P_{i,j^*}|$ time and argue that

$$
\begin{aligned}
\text{cost}(\mathcal{C}, P_{i,j}) &= \sum_{x \in P_{i,j}} \text{cost}(\mathcal{C}, x) \\
&\leq \sum_{x \in P_{i,j}} \text{cost}(\mathcal{C}, y) + 2C_0 \cdot \text{diam}(P_{i,j}) \\
&\leq \eta \cdot \text{cost}(\mathcal{C}, P_{i,j^*}) + 2C_0 \cdot |P_{i,j}| \cdot \text{diam}(P_{i,j}),
\end{aligned}
$$

as desired. $\square$

Using Lemma B.13 and Lemma B.14, we are now ready to bound the total cost on rings that are marked "*processed*" without any point being added to $\mathsf{SC}$ as follows.

**Lemma B.15.** *Let $\mathcal{P}^{\text{light}}$ be the set of rings that are marked "*processed*" without points being added to $\mathsf{SC}$, and let $\mathcal{P}^{\text{heavy}}$ be all other rings. For any set of centers $\mathcal{C}$, we have that*

$$
\text{cost}(\mathcal{C}, \mathcal{P}^{\text{light}}) := \sum_{P_{i,j} \in \mathcal{P}^{\text{light}}} \text{cost}(\mathcal{C}, P_{i,j}) \leq \varepsilon \cdot 2C_0^2 \cdot \log n \cdot \sum_{P_{i,j} \in \mathcal{P}^{\text{heavy}}} \left( \text{cost}(\mathcal{C}, P_{i,j}) + |P_{i,j}| \cdot \text{diam}(P_{i,j}) \right).
$$

*Proof.* The lemma is a natural corollary of Lemma B.13 and Lemma B.14. Fix any ring $P_{i,j^*} \in \mathcal{P}^{\text{heavy}}$, we let the rings $\mathcal{P}^{\text{light}}(j^*)$ be the set of rings with index $\ell$ such that $j^*(\ell) = j^*$ (the notation was first defined before Algorithm 6). We apply Lemma B.14 and Lemma B.13 to all rings in $\mathcal{P}^{\text{light}}(j^*)$ and obtain that

$$
\begin{aligned}
\sum_{P_{i,j} \in \mathcal{P}^{\text{light}}(j^*)} \text{cost}(\mathcal{C}, P_{i,j}) &\leq \sum_{P_{i,j} \in \mathcal{P}^{\text{light}}(j^*)} \frac{\varepsilon}{2C_0} \cdot \text{cost}(\mathcal{C}, P_{i,j^*}) + 2C_0 \cdot \sum_{P_{i,j} \in \mathcal{P}^{\text{light}}(j^*)} |P_{i,j}| \cdot \text{diam}(P_{i,j}) && \text{(by Lemma B.14)} \\
&\leq \sum_{P_{i,j} \in \mathcal{P}^{\text{light}}(j^*)} \frac{\varepsilon}{2C_0} \cdot \text{cost}(\mathcal{C}, P_{i,j^*}) + \varepsilon \cdot 2C_0^2 \cdot \sum_{P_{i,j} \in \mathcal{P}^{\text{light}}(j^*)} |P_{i,j^*}| \cdot \text{diam}(P_{i,j^*}) \\
& && \text{(by Lemma B.13)} \\
&\leq \log n \cdot \left( \frac{\varepsilon}{2C_0} \cdot \text{cost}(\mathcal{C}, P_{i,j^*}) + \varepsilon \cdot 2C_0^2 \cdot |P_{i,j^*}| \cdot \text{diam}(P_{i,j^*}) \right) && \text{(at most } \log n \text{ such rings)} \\
&\leq \varepsilon \cdot 2C_0^2 \cdot \log n \cdot \left( \text{cost}(\mathcal{C}, P_{i,j^*}) + |P_{i,j^*}| \cdot \text{diam}(P_{i,j^*}) \right).
\end{aligned}
$$

Finally, since each ring $P_{i,\ell}$ in $\mathcal{P}^{\text{light}}$ has a unique $j^*(\ell)$, summing over all rings in $\mathcal{P}^{\text{light}}$ gives the desired lemma statement. $\square$

**Wrapping up the proof of Theorem 4.** We are now ready to finalize the proof of Theorem 4. The following claim is a natural corollary of the $\beta$-approximation of the $(\alpha, \beta)$-weak coreset.

**Claim B.16.** *Let OPT be the optimal cost of clustering on $P$ with $k$ centers and let $\mathcal{A}$ be the set of centers from a $\beta$-approximate solution. Then,*

$$
\sum_{i,j} |P_{i,j}| (2C_0)^j R \leq 3C_0 \beta \cdot \text{OPT} \qquad \& \qquad \sum_{i,j} |P_{i,j}| \text{diam}(P_{i,j}) \leq 6C_0 \beta \cdot \text{OPT}
$$

.

*Proof.* Consider any point $p \in P_{i,j}$. For $j = 0$, $(2C_0)^j R = R$ and for $j \geq 1$, $(2C_0)^j R \leq 2C_0 \mathbf{d}(\mathcal{A}, p)$. Hence, for any $j$ we have $(2C_0)^j R \leq 2C_0 \mathbf{d}(\mathcal{A}, p) + R$. Now,

$$
\sum_{i,j} |P_{i,j}|(2C_0)^j R = \sum_{i,j} \sum_{p \in P_{i,j}} (2C_0)^j R \leq \sum_{i,j} \sum_{p \in P_{i,j}} 2C_0 \mathbf{d}(\mathcal{A}, p) + R
$$
$$
= \sum_{p \in X} 2C_0 \mathbf{d}(\mathcal{A}, p) + R
$$
$$
\leq 2C_0 \beta \text{OPT} + nR \leq 3C_0 \beta \cdot \text{OPT}
$$

Since $\text{diam}(P_{i,j}) \leq 2(2C_0)^j R$, we have $\sum_{i,j} |P_{i,j}| \text{diam}(P_{i,j}) \leq 6C_0 \beta \cdot \text{OPT}$

$\square$

**Lemma B.17.** *For any set of centers $\mathcal{C} \subseteq X$ of size at most $k$, it holds that*

$$
|\text{cost}(\mathcal{C}, X) - \text{cost}(\mathcal{C}, \mathsf{SC})| \leq \varepsilon \cdot 15\beta \cdot C_0^3 \cdot \log n \cdot \text{cost}(\mathcal{C}, X)
$$

*with probability at least $1 - 1/\text{poly}(n)$.*

*Proof.* Similar to the case in Lemma B.9 and Lemma B.15, we define $\mathcal{P}^{\text{light}}$ as the rings being marked as "*processed*" without points added to $\mathsf{SC}$ and $\mathcal{P}^{\text{heavy}}$ as the other rings. We thus have

$$
\text{cost}(\mathcal{C}, X) = \sum_{P_{i,j}^{\text{light}} \in \mathcal{P}^{\text{light}}} \text{cost}(\mathcal{C}, P_{i,j}^{\text{light}}) + \sum_{P_{i,j}^{\text{heavy}} \in \mathcal{P}^{\text{light}}} \text{cost}(\mathcal{C}, P_{i,j}^{\text{heavy}}).
$$

Define $\widetilde{\text{cost}}(\mathcal{C}, \mathcal{P}^{\text{light}})$ and $\widetilde{\text{cost}}(\mathcal{C}, \mathcal{P}^{\text{heavy}})$ as the cost induced by $\mathsf{SC}$ for $\mathcal{C}$. Clearly, the total cost induced by the coreset $\mathsf{SC}$ is $\text{cost}(\mathcal{C}, \mathsf{SC}) = \widetilde{\text{cost}}(\mathcal{C}, \mathcal{P}^{\text{light}}) + \widetilde{\text{cost}}(\mathcal{C}, \mathcal{P}^{\text{heavy}})$. Furthermore, by our construction, there is $\widetilde{\text{cost}}(\mathcal{C}, \mathcal{P}^{\text{light}}) = 0$. Conditioning on the high-probability events of Lemma B.9 and Lemma B.15, we have that

$$
\left|\text{cost}(\mathcal{C}, \mathcal{P}^{\text{heavy}}) - \widetilde{\text{cost}}(\mathcal{C}, \mathcal{P}^{\text{heavy}})\right| \leq 2 \cdot \varepsilon \cdot \sum_{P_{i,j} \in \mathcal{P}^{\text{heavy}}} |P_{i,j}| \text{diam}(P_{i,j})
$$
$$
\left|\text{cost}(\mathcal{C}, \mathcal{P}^{\text{light}}) - \widetilde{\text{cost}}(\mathcal{C}, \mathcal{P}^{\text{light}})\right| \leq \text{cost}(\mathcal{C}, \mathcal{P}^{\text{light}})
$$
$$
\leq \varepsilon \cdot 2C_0^2 \cdot \log n \cdot \sum_{P_{i,j} \in \mathcal{P}^{\text{heavy}}} \left(\text{cost}(\mathcal{C}, P_{i,j}) + |P_{i,j}| \cdot \text{diam}(P_{i,j})\right).
$$

As such, we could bound the difference of the cost as

$$
|\text{cost}(\mathcal{C}, X) - \text{cost}(\mathcal{C}, \mathsf{SC})|
$$
$$
\leq \left|\text{cost}(\mathcal{C}, \mathcal{P}^{\text{heavy}}) - \widetilde{\text{cost}}(\mathcal{C}, \mathcal{P}^{\text{heavy}})\right| + \left|\text{cost}(\mathcal{C}, \mathcal{P}^{\text{light}}) - \widetilde{\text{cost}}(\mathcal{C}, \mathcal{P}^{\text{light}})\right|
$$
$$
\leq \varepsilon \cdot 2C_0^2 \cdot \log n \cdot \sum_{P_{i,j} \in \mathcal{P}^{\text{heavy}}} |P_{i,j}| \cdot \text{diam}(P_{i,j}) + \varepsilon \cdot 2C_0^2 \cdot \log n \cdot \sum_{P_{i,j} \in \mathcal{P}^{\text{heavy}}} \text{cost}(\mathcal{C}, P_{i,j}) \quad \text{(by Lemma B.15)}
$$
$$
\leq \varepsilon \cdot \beta \cdot 12C_0^3 \log n \cdot \text{OPT} + \varepsilon \cdot 2C_0^2 \cdot \log n \cdot \sum_{P_{i,j} \in \mathcal{P}^{\text{heavy}}} \text{cost}(\mathcal{C}, P_{i,j}) \quad \text{(by Claim B.16)}
$$
$$
\leq \varepsilon \cdot \beta \cdot 12C_0^3 \log n \cdot \text{cost}(\mathcal{C}, X) + \varepsilon \cdot 2C_0^2 \cdot \log n \cdot \text{cost}(\mathcal{C}, X)
$$
$$
\leq \varepsilon \cdot 15\beta \cdot C_0^3 \cdot \log n \cdot \text{cost}(\mathcal{C}, X) \quad (\beta \geq 1)
$$

which is as desired. $\square$

Finally, to get $(1 + \varepsilon)$-approximation, we let $\varepsilon = \frac{\varepsilon'}{15\beta \cdot \varepsilon \cdot C_0^3 \cdot \log n} = O(\frac{\varepsilon'}{\log n})$ by the constant choices of $\beta$ and $C_0$ (by Propositions 2 and 3). The size of the coreset and number of $\mathsf{SO}$ queries are therefore $O(k^2 \log^5 n/\varepsilon'^3) = O(k^2 \log^8 n/\varepsilon^3) = k \text{ polylog}(n)/\varepsilon^3$, as desired by Theorem 4.

# C    Deferred Proofs from Section 4

**The analysis.**    Similar to the analysis of Algorithm 1, we need to show that $i$). Algorithm 5 converges with a small number of strong oracle queries; $ii$). Algorithm 5 preserves the $k$-median cost by a $(1 + \varepsilon)$ factor. In addition, we need to prove that Algorithm 5 is indeed assignment-preserving.

We first establish the bounded number of strong oracle queries for Algorithm 5. The guarantee and the proof are as follows.

**Lemma C.1.** *Algorithm 5 outputs a coreset of size $O(\frac{k^2 \log^4 n \log n/\varepsilon}{\varepsilon^2})$ using $O(k \log^4 n)$ strong oracle* SO *(point or edge) queries, $O(nk \log^3 n)$ weak oracle* WO *queries, and converges in $\widetilde{O}(k^2/\varepsilon^2 + nk)$ time. Furthermore, with probability at least $1 - 1/\operatorname{poly}(n)$, after $10 \log^2 n$ iteration of line 4c, all point of $P_i$ is processed by* exactly one *of the iterations of line 4(c)v.*

*Proof.* For each iteration, the only points we make strong oracle SO queries on are the points in $S_r$, which is at most $O(\log^2 n)$. All other points in $P_i$ make at most $O(\log n)$ times of weak oracle queries, and the number of points added to FC during each iteration is $\Theta(k\varepsilon^{-2} \log^2 n \log(n\varepsilon^{-1}))$.

We now prove the "furthermore" part using the high-probability events of Lemma B.3, Lemma B.4, and Claim B.5. Note that although the choice of $j^*$ and the number of samples have changed, we still have $\left| P_{i,j^*} \cap \widetilde{P}_i \right| \geq 800C_0 \cdot \log n$, which means the guarantees in Claim B.5 continues to hold with a changed constant. Therefore, each point is only processed by one iteration, and in each iteration, at least $\frac{1}{50 \log n}$ fraction of the remaining points are processed by line 4(c)v. This implies after $100 \log^2 n$ iterations, no point in $P_i$ remains not processed by line 4(c)v.

Therefore, the size of the coreset FC follows from the number of $O(k\varepsilon^{-2} \log^2 n \log(n\varepsilon^{-1}))$ samples in each iteration and the $k \log^2 n$ iterations. The number of weak and strong oracle queries follows from the same argument, and the running time scales linearly with the coreset size. $\qquad \square$

We now move to the analysis of the assignment-preserving $(1 + \varepsilon)$ coreset. The main lemma for the assignment-preserving approximation is as follows.

**Lemma C.2.** *Let $(\mathcal{X}, d)$ be an input set of points in $\mathbb{R}^d$, and let* FC *be the resulting assignment-preserving coreset as prescribed by Algorithm 5. With high probability, for any set of centers $\mathcal{C}$ and any given assignment constraint $\Gamma$, there is*

$$|\operatorname{cost}(\mathsf{FC}, \mathcal{C}, \Gamma) - \operatorname{cost}(\mathcal{X}, \mathcal{C}, \Gamma)| \leq \varepsilon \cdot \operatorname{cost}(\mathcal{X}, \mathcal{C}, \Gamma).$$

To prove Lemma C.2, we first reduce the problem to an easier case that $\mathcal{C}, \Gamma$ is fixed, using a union bound similar to Braverman et al. (2022). We also make diam$(\mathcal{C})$ bounded by moving centers far from the ring center. Next we bound the additive error for a *single* set of points $T_i^{(r)}$ and $\mathsf{FC}(T_i^{(r)})$ (recall that $T_i^{(r)}$ is the set of peeled points and $\mathsf{FC}(T_i^{(r)})$ is weighted sampled set from $T_i^{(r)}$). We construct transformations between assignments of $T_i^{(r)}$ and $\mathsf{FC}(T_i^{(r)})$ that suffers small cost. These transformations imply optimal costs of $T_i^{(r)}$ and $\mathsf{FC}(T_i^{(r)})$ are close.

For the clarity of presentation, in what follows, we use $T$ and $c^*$ as short-hand notations for $T_i^{(r)}$ and $c_i$ when the context is clear. Let $L$ be the number of rings of $c^*$.

**Step I: Union bound.**    Our first step is reducing Lemma C.2 to a much easier version, in which $\mathcal{C}$ and $\Gamma$ are fixed and $\mathcal{C}$ does not contain centers far from $c^*$. In particular, we move all centers outside $B(c^*, R_{\text{far}})$ to $c^*$, where $R_{\text{far}} = 60\text{diam}(T)/\varepsilon$.

Let the approximation ratio of WC be $\alpha_{\mathsf{WC}}$. Define $\varepsilon' = \frac{\varepsilon}{10C_0^2 \alpha_{\mathsf{WC}} \log n}$. The proof of Lemma C.2 requires the additive error to be less than $\frac{\varepsilon}{10} \cdot \operatorname{cost}(T, C, \Gamma) + \varepsilon' \cdot |T| \cdot \operatorname{diam}(T)$.

**Lemma C.3.** *Suppose $\operatorname{ALG}(T)$ is an algorithm that outputs assignment-preserving coresets of $T$. If for every $T \subseteq \mathcal{X}, \tilde{\mathcal{C}} \subset B(c^*, 60\text{diam}(T)/\varepsilon)$ of size $k$ and every assignment constraint $\tilde{\Gamma}$,*

$$\Pr \left[ \left| \operatorname{cost}(\operatorname{ALG}(T), \tilde{\mathcal{C}}, \tilde{\Gamma}) - \operatorname{cost}(T, \tilde{\mathcal{C}}, \tilde{\Gamma}) \right| \geq \frac{10}{11} \varepsilon' \cdot |T| \cdot \operatorname{diam}(T) \right] \leq \delta$$

*for some $\delta \in (0, 1)$, then for every $T \subseteq \mathcal{X}$, with probability at least $1 - O(n^k \cdot (n + k\varepsilon^{-1})^k \delta)$, for every center set $\mathcal{C} \subset \mathcal{X}$ of size $k$ and every assignment constraint $\Gamma$,*

$$|\mathrm{cost}(\mathrm{ALG}(T), \mathcal{C}, \Gamma) - \mathrm{cost}(T, \mathcal{C}, \Gamma)| \leq \frac{\varepsilon}{10} \cdot \mathrm{cost}(T, \mathcal{C}, \Gamma) + \varepsilon' \cdot |T| \cdot \mathrm{diam}(T).$$

*Proof.* For every center set $\mathcal{C}$ and assignment constraint $\Gamma$, we move all centers outside $B(c^*, R_{\mathrm{far}})$ (denoted by $\mathcal{C}_{\mathrm{far}}$) to $c^*$ and define the center set and assignment constraint after movements as $\tilde{\mathcal{C}}$ and $\tilde{\Gamma}$. For every assignment $\sigma$, the cost of $\sigma$ before movements is upper bounded by

$$\mathrm{cost}^\sigma(T, \mathcal{C}, \Gamma) = \mathrm{cost}^\sigma(T, \tilde{\mathcal{C}}, \tilde{\Gamma}) + \sum_{x \in T} \sum_{c \in \mathcal{C}_{\mathrm{far}}} \sigma(x, c)(\mathbf{d}(x, c) - \mathbf{d}(x, c^*))$$

$$\leq \mathrm{cost}^\sigma(T, \tilde{\mathcal{C}}, \tilde{\Gamma}) + \sum_{x \in T} \sum_{c \in \mathcal{C}_{\mathrm{far}}} \sigma(x, c)\mathbf{d}(c, c^*)$$

$$= \mathrm{cost}^\sigma(T, \tilde{\mathcal{C}}, \tilde{\Gamma}) + \sum_{c \in \mathcal{C}_{\mathrm{far}}} \Gamma(c)\mathbf{d}(c, c^*),$$

and lower bounded by

$$\mathrm{cost}^\sigma(T, \mathcal{C}, \Gamma) = \mathrm{cost}^\sigma(T, \tilde{\mathcal{C}}, \tilde{\Gamma}) + \sum_{x \in T} \sum_{c \in \mathcal{C}_{\mathrm{far}}} \sigma(x, c)(\mathbf{d}(x, c) - \mathbf{d}(x, c^*))$$

$$\geq \mathrm{cost}^\sigma(T, \tilde{\mathcal{C}}, \tilde{\Gamma}) + \sum_{x \in T} \sum_{c \in \mathcal{C}_{\mathrm{far}}} \sigma(x, c)(\mathbf{d}(c, c^*) - 2\mathbf{d}(x, c^*))$$

$$\geq \mathrm{cost}^\sigma(T, \tilde{\mathcal{C}}, \tilde{\Gamma}) + \sum_{x \in T} \sum_{c \in \mathcal{C}_{\mathrm{far}}} (1 - \varepsilon/20)\sigma(x, c)\mathbf{d}(c, c^*)$$

$$= \mathrm{cost}^\sigma(T, \tilde{\mathcal{C}}, \tilde{\Gamma}) + (1 - \varepsilon/20) \sum_{c \in \mathcal{C}_{\mathrm{far}}} \Gamma(c)\mathbf{d}(c, c^*).$$

In corollary, suppose $\sigma^*$ and $\widetilde{\sigma}^*$ are optimal assignments of $\mathrm{cost}(T, \mathcal{C}, \Gamma)$ and $\mathrm{cost}(T, \tilde{\mathcal{C}}, \tilde{\Gamma})$, then we have

$$\mathrm{cost}(T, \tilde{\mathcal{C}}, \tilde{\Gamma}) \leq \mathrm{cost}^{\sigma^*}(T, \tilde{\mathcal{C}}, \tilde{\Gamma}) \leq \mathrm{cost}(T, \mathcal{C}, \Gamma) - (1 - \varepsilon/20) \sum_{c \in \mathcal{C}_{\mathrm{far}}} \Gamma(c)\mathbf{d}(c, c^*)$$

$$\mathrm{cost}(T, \mathcal{C}, \Gamma) \leq \mathrm{cost}^{\widetilde{\sigma}^*}(T, \mathcal{C}, \Gamma) \leq \mathrm{cost}(T, \tilde{\mathcal{C}}, \tilde{\Gamma}) + \sum_{c \in \mathcal{C}_{\mathrm{far}}} \Gamma(c)\mathbf{d}(c, c^*).$$

Next we round all capacities in $\tilde{\Gamma}$ to $\{i/M : i \geq 0\}$, where $M = \lceil 600k/\varepsilon \rceil$, to get a new assignment constraint $\tilde{\Gamma}'$ satisfying $\tilde{\Gamma}(\mathcal{C}) = \tilde{\Gamma}'(\mathcal{C})$ and $|\tilde{\Gamma}(c) - \tilde{\Gamma}'(c)| < 1/M$ for every $c \in \mathcal{C}$.

Suppose the optimal assignment for $\tilde{\Gamma}$ is $\sigma$. There always exists an assignment $\sigma'$ of $\tilde{\Gamma}'$ such that for every $c \in \tilde{\mathcal{C}}$, $|\sum_{x \in T}(\sigma(x, c) - \sigma'(x, c))| \leq 1/M$. Hence we have

$$|\mathrm{cost}(T, \tilde{\mathcal{C}}, \tilde{\Gamma}) - \mathrm{cost}(T, \tilde{\mathcal{C}}, \tilde{\Gamma}')| \leq \sum_{c \in \tilde{\mathcal{C}}} \left| \sum_{x \in T} \sigma(x, c)\mathbf{d}(x, c) - \sum_{x \in T} \sigma'(x, c)\mathbf{d}(x, c) \right|$$

$$\leq \sum_{c \in \tilde{\mathcal{C}}} \frac{\mathrm{diam}(\tilde{\mathcal{C}})}{M}$$

$$\leq \frac{60k \cdot \mathrm{diam}(T)}{\varepsilon M}$$

$$\leq \mathrm{diam}(T)/10.$$

Define

$$\mathcal{F} := \{(\mathcal{C}, \Gamma) : \mathcal{C} \subset B(c^*, R_{\mathrm{far}}), |\mathcal{C}| = k, \Gamma(\mathcal{C}) = |T|, \forall c \in \mathcal{C}, \Gamma(c) \in \{i/M : i > 0\}\}.$$

We have $|\mathcal{F}| \leq \Delta^{kd}(|T| + M)^k$. Applying union bound for every $(\tilde{\Gamma}', \tilde{\mathcal{C}}) \in \mathcal{F}$, we have

$$\forall (\tilde{\Gamma}', \tilde{\mathcal{C}}) \in \mathcal{F}, |\text{cost}(S, \tilde{\mathcal{C}}, \tilde{\Gamma}') - \text{cost}(T, \tilde{\mathcal{C}}, \tilde{\Gamma}')| \leq \frac{\varepsilon}{11C_0 \log n} \cdot |T| \cdot \text{diam}(T) \tag{2}$$

holds with probability at least $1 - |\mathcal{F}|\delta$. Conditioning on (2), for every $\mathcal{C}$ and $\Gamma$, we have

$$
\begin{aligned}
& |\text{cost}(S, \mathcal{C}, \Gamma) - \text{cost}(T, \mathcal{C}, \Gamma)| \\
\leq{}& |(\text{cost}(S, \mathcal{C}, \Gamma) - \text{cost}(S, \tilde{\mathcal{C}}, \tilde{\Gamma})) + (\text{cost}(T, \mathcal{C}, \Gamma) - \text{cost}(T, \tilde{\mathcal{C}}, \tilde{\Gamma}))| + |\text{cost}(T, \tilde{\mathcal{C}}, \tilde{\Gamma}) - \text{cost}(S, \tilde{\mathcal{C}}, \tilde{\Gamma})| \\
\leq{}& \frac{\varepsilon}{10} \sum_{c \in \mathcal{C}_{\text{far}}} \Gamma(c)\mathbf{d}(c, c^*) + |\text{cost}(S, \tilde{\mathcal{C}}, \tilde{\Gamma}') - \text{cost}(S, \tilde{\mathcal{C}}, \tilde{\Gamma})| + |\text{cost}(T, \tilde{\mathcal{C}}, \tilde{\Gamma}') - \text{cost}(T, \tilde{\mathcal{C}}, \tilde{\Gamma})| \\
& + |\text{cost}(T, \tilde{\mathcal{C}}, \tilde{\Gamma}') - \text{cost}(S, \tilde{\mathcal{C}}, \tilde{\Gamma}')| \\
\leq{}& \frac{\varepsilon}{10} \sum_{c \in \mathcal{C}_{\text{far}}} \Gamma(c)\mathbf{d}(c, c^*) + \frac{1}{10}\text{diam}(T) + \frac{10}{11}\varepsilon' \cdot |T| \cdot \text{diam}(T) \\
\leq{}& \frac{\varepsilon}{10}\text{cost}(T, \mathcal{C}, \Gamma) + \varepsilon' \cdot |T| \cdot \text{diam}(T).
\end{aligned}
$$

$\square$

**Step II: Bounding** $|\text{cost}(\mathsf{FC}(T), \mathcal{C}, \Gamma) - \text{cost}(T, \mathcal{C}, \Gamma)|$**.** In order to prove $\text{cost}(T, \mathcal{C}, \Gamma)$ and $\text{cost}(\mathsf{FC}(T), \mathcal{C}, \Gamma)$ are close, we show that optimal assignments of $T$ and $\mathsf{FC}(T)$ can transform to each other. Suppose $\sigma^*, \tilde{\sigma}^*$ are optimal assignments of $T$ and $\mathsf{FC}(T)$ respectively.

Fix any set of points $T$, centers $\mathcal{C}$, and capacity constraint $\Gamma$. We will prove the case for $\text{diam}(\mathcal{C}) \leq R_{\text{far}} = 60\text{diam}(T)/\varepsilon$ and apply the union bound by Lemma C.3 afterwards.

Starting with $\tilde{\sigma}^*$, our first target is to construct an assignment $\sigma$ of $T$ such that $\text{cost}^\sigma(T, \mathcal{C}, \Gamma) \approx \text{cost}^{\tilde{\sigma}^*}(\mathsf{FC}(T), \mathcal{C}, \Gamma)$.

**Lemma C.4.** *For any fixed choice of $\mathcal{C}$, $\Gamma$, and $T$ for some iteration $r$ and $i$ of line 4(c)v, we have that*

$$\text{cost}(T, \mathcal{C}, \Gamma) \leq \mathbb{E}\left[\text{cost}(\mathsf{FC}(T), \mathcal{C}, \Gamma)\right].$$

*The expectation is taken over only the randomness of the sampling in line 4(c)vC.*

*Proof.* Let $\tilde{\sigma}^*$ be the optimal assignment of $\mathsf{FC}(T)$. Since $\tilde{\sigma}^*$ is a random assignment, we construct $\sigma$ by setting $\sigma(x, c) := \mathbb{E}\left[\tilde{\sigma}^*(x, c)\right]$. Then we have

$$
\begin{aligned}
\mathbb{E}\left[\text{cost}(\mathsf{FC}(T), \mathcal{C}, \Gamma)\right] &= \mathbb{E}\left[\sum_{c \in \mathcal{C}} \sum_{x \in T} \tilde{\sigma}(x, c)\right] && \text{(by definition)} \\
&= \sum_{c \in \mathcal{C}} \sum_{x \in T} \sigma(x, c) && \\
&= \text{cost}^\sigma(T, \mathcal{C}, \Gamma) && \text{(by definition)} \\
&\geq \text{cost}^{\sigma^*}(T, \mathcal{C}, \Gamma), && \text{(optimality of } \sigma^*)
\end{aligned}
$$

as desired. $\square$

**Lemma C.5.** *For any fixed choice of $\mathcal{C}$, $\Gamma$, and $T$ for some iteration $r$ and $i$ of line 4(c)v, with probability at least $1 - n^{-10} \cdot O(n^{-k}(n + k\varepsilon^{-1})^{-k})$, we have*

$$\text{cost}(\mathsf{FC}(T), \mathcal{C}, \Gamma) \leq \text{cost}(T, \mathcal{C}, \Gamma) + \varepsilon' \cdot |T| \cdot \text{diam}(T).$$

*The probability is taken over only the randomness of the sampling in line 4(c)vC.*

*Proof.* Let $\sigma^*$ be the optimal assignment of $T$. We start with an assignment that assigns $\widetilde{\sigma}'(x,c) = \sigma^*(x,c)\frac{|T|}{m}$ fraction of $x$ to center $c$. This assignment satisfies $\forall x \in \mathsf{FC}(T), \sum_{c \in \mathcal{C}} \widetilde{\sigma}'(x,c) = \frac{|T|}{m}$. However, $\sum_{x \in T} \widetilde{\sigma}'(x,c) = \Gamma(c)$ may not hold for every $c \in \mathcal{C}$.

In order to transform $\widetilde{\sigma}'(x,c)$ into an assignment satisfying $\Gamma$, for every center such that $\sum_{x \in T} \widetilde{\sigma}'(x,c) > \Gamma(c)$, we need to change some assignments of $c$ to other centers $x \to c'$, suffering cost of at most $\mathbf{d}(x,c') - \mathbf{d}(x,c) \le \mathbf{d}(c,c') \le \mathrm{diam}(\mathcal{C})$.

For every center $c \in \mathcal{C}$, the difference between assigned weight and real capacity is $|\sum_{x \in \mathsf{FC}(T)} \sigma(x,c)\frac{|T|}{m} - \Gamma(c)|$. For every $T$, we have

$$
\mathrm{cost}(\mathsf{FC}(T), \mathcal{C}, \Gamma) \le \sum_{c \in \mathcal{C}} \sum_{x \in T} \widetilde{\sigma}(x,c)
$$

$$
\le \sum_{c \in \mathcal{C}} \sum_{x \in T} \widetilde{\sigma}'(x,c) + \mathrm{diam}(\mathcal{C}) \sum_{c \in \mathcal{C}} \left| \sum_{x \in \mathsf{FC}(T)} \sigma(x,c)\frac{|T|}{m} - \Gamma(c) \right|
$$

$$
= \sum_{c \in \mathcal{C}} \sum_{x \in T} \sigma(x,c) + \mathrm{diam}(\mathcal{C}) \sum_{c \in \mathcal{C}} \left| \sum_{x \in \mathsf{FC}(T)} \sigma(x,c)\frac{|T|}{m} - \Gamma(c) \right|
$$

$$
= \sum_{c \in \mathcal{C}} \sum_{x \in T} \sigma(x,c) + \mathrm{diam}(\mathcal{C}) \max_{s \in \{-1,+1\}^{\mathcal{C}}} \sum_{c \in \mathcal{C}} s_c \left( \sum_{x \in \mathsf{FC}(T)} \sigma(x,c)\frac{|T|}{m} - \Gamma(c) \right)
$$

Fix $s \in \{-1,+1\}^{\mathcal{C}}$, let $x_i$ ($1 \le i \le m$) denotes the $i$-th point in $T$. Then $x_1, x_2, \ldots, x_m$ are independent random variables uniformly drawn from $T$. Define $Y_i = \sum_{c \in \mathcal{C}} s_c \sigma(x_i,c)\frac{|T|}{m}$. For every $1 \le i \le m$, we have $Y_i \in [-|T|/m, |T|/m]$. By Hoeffding's inequality,

$$
\Pr\left[ \sum_{i=1}^{m} \sum_{c \in \mathcal{C}} s_c \sigma(x_i,c)\frac{|T|}{m} - \sum_{c \in \mathcal{C}} s_c \Gamma(c) > \varepsilon'T \right] = \Pr\left[ \sum_{i=1}^{m} Y_i - \mathbb{E}\left[ \sum_{i=1}^{m} Y_i \right] > \varepsilon'T \right]
$$
$$
\le \exp\left( -\varepsilon'^2 m/2 \right).
$$

Applying union bound for every $s \in \{-1,+1\}^{\mathcal{C}}$, we have

$$
\Pr\left[ \max_{s \in \{-1,+1\}^{\mathcal{C}}} \sum_{c \in \mathcal{C}} s_c \left( \sum_{x \in \mathsf{FC}(T)} \sigma(x,c)\frac{|T|}{m} - \Gamma(c) \right) > \varepsilon'T/2 \right] \le 2^k \exp\left( -\varepsilon'^2 m/2 \right)
$$
$$
\le n^{-10} \cdot O(n^{-k}(n + k\varepsilon^{-1})^{-k}).
$$

In summary, $\mathrm{cost}(\mathsf{FC}(T), \mathcal{C}, \Gamma) \le \mathrm{cost}(T, \mathcal{C}, \Gamma) + \varepsilon' \cdot |T| \cdot \mathrm{diam}(T)$ holds with probability at least $1 - n^{-10} \cdot O(n^{-k}(n + k\varepsilon^{-1})^{-k})$. $\qquad\square$

We now move to bound the error induced by $\mathrm{cost}(\mathsf{FC}(T), \mathcal{C}, \Gamma) - \mathbb{E}\left[\mathrm{cost}(\mathsf{FC}(T), \mathcal{C}, \Gamma)\right]$ in the same manner of Cohen-Addad & Li (2019); Cohen-Addad et al. (2025).

**Lemma C.6.** *For any fixed choice of $\mathcal{C}$, $\Gamma$, and $T$ for some iteration $r$ and $i$ of line 4(c)v, with probability at least $1 - n^{-10} \cdot O(n^{-k}(n + k\varepsilon^{-1})^{-k})$, we have that*

$$
|\mathrm{cost}(\mathsf{FC}(T), \mathcal{C}, \Gamma) - \mathbb{E}\left[\mathrm{cost}(\mathsf{FC}(T), \mathcal{C}, \Gamma)\right]| \le \varepsilon' \cdot |T| \cdot \mathrm{diam}(T).
$$

*The randomness is only over the sampling in line 4(c)vC.*

*Proof.* Let $t_1, t_2, \ldots, t_m$ be the $m$ points sampled in $\mathsf{FC}(T)$. Define $f(t_1, t_2, \ldots, t_m) = \mathrm{cost}(\{t_1, \ldots, t_m\}, \mathcal{C}, \Gamma)$. For every $\mathbf{t}, \in \mathcal{X}^m$ and $1 \le i \le m$, we have

$$
\sup_{t_i' \in \mathcal{X}} |f(t_1, \ldots, t_i, \ldots, t_m) - f(t_1, \ldots, t_i', \ldots, t_m)| \le \frac{|T|}{m} \mathrm{diam}(T).
$$

By McDiarmid's inequality,

$$\Pr[|f(t_1, \ldots, t_m) - \mathbb{E}\left[f(t_1, \ldots, t_m)\right]| \leq \varepsilon \cdot |T| \cdot \operatorname{diam}(T)] \leq \exp(-\varepsilon'^2 m)$$
$$\leq n^{-10} \cdot O(n^{-k}(n + k\varepsilon^{-1})^{-k}).$$

$\square$

**Step III: Put everything together.** We finalize the proof of Lemma C.2. To this end, we need to further establish a lower bound for the cost of $\operatorname{cost}(T_i^{(r)}, \mathcal{C}, \Gamma_i^{(r)})$ for each iteration $r$ as follows. In the following discussion, $\operatorname{cost}(\mathcal{X}, \mathsf{WC})$ and $\operatorname{cost}(\mathcal{X})$ denotes the optimal cost of unconstrained $k$-median problem. In particular $\operatorname{cost}(\mathcal{X})$ denotes $\min_{\mathcal{C} \subseteq \mathcal{X}, |\mathcal{C}| = k} \operatorname{cost}(\mathcal{X}, \mathcal{C})$.

**Lemma C.7.** *With probability at least $1 - n^{-7}$,*

$$\operatorname{cost}(\mathcal{X}, \mathsf{WC}) \geq \frac{1}{8C_0^2 \log n} \sum_{i=1}^{|\mathsf{WC}|} \sum_{r=1}^{10 \log n} \left|T_i^{(r)}\right| \operatorname{diam}(T_i^{(r)}). \tag{3}$$

*Proof.* Let $P_{i,j}^{(r)}$ be the remaining points at the beginning of the $r$-th iteration of line 4c. Let $N_r = \sum_j \left|P_{i,j}^{(r)}\right|$. We fix some $r$ in the following discussion and omit the superscript $(r)$ for convenience.

Let $j^*$ follows the definition in line 4(c)iv. Define $\gamma = \frac{4}{5 \log n} s_r$. We have $|S_{i,j^*}| \geq \gamma$ by the definition of $j^*$. Define $s_{i,j} = \frac{|P_{i,j}|}{N} s_r$ be the expected size of $S_{i,j}$.

Pick $\delta = 0.1$. By Chernoff bound, we have

$$\Pr\left[|P_{i,j^*}| \leq \frac{4}{5(1+\delta) \log n} N\right] = \Pr\left[(1+\delta)s_{i,j^*} \leq \gamma\right]$$
$$\leq \Pr\left[(1+\delta)s_{i,j^*} \leq |S_{i,j^*}|\right]$$

When $s_{i,j^*} \geq \frac{\gamma}{1+\delta}$, we have

$$\Pr\left[|S_{i,j^*}| \geq (1+\delta)s_{i,j^*}\right] \leq \exp\left(-\frac{\delta^2}{(\delta+2)(1-\delta)}\gamma\right) \qquad \text{(Chernoff bound)}$$
$$\leq n^{-10} \qquad (\gamma = \Omega(\log n))$$

Otherwise, we have $|S_{i,j^*}| > (\gamma/s_{i,j^*})s_{i,j^*}$. Since $\gamma/s_{i,j^*}$ is sufficiently large, this case happens with negligible probability. We show that with high probability, for every $1 \leq j \leq L$ satisfying $s_{i,j} < \frac{\gamma}{1+\delta}$, $|S_{i,j^*}| \leq \gamma$ holds.

$$\Pr[|S_{i,j}| \geq \gamma] \leq \exp\left(-\frac{t^2}{t+2}s_{i,j}\right) \qquad (t = \frac{\gamma}{s_{i,j}} - 1)$$
$$\leq \exp\left(-\frac{t\delta}{(t+2)(1+\delta)}\gamma\right) \qquad (s_{i,j} < \frac{\gamma}{1+\delta})$$
$$\leq n^{-11}. \qquad (\gamma = \Omega(\log n))$$

Take union bound for every $j$, we have $\Pr[s_{i,j^*} \geq \frac{\gamma}{1+\delta}] \geq 1 - n^{-10}$.

In summary, with probability at least $1 - 2n^{-10}$,

$$|P_{i,j^*}| \geq \frac{4}{5(1+\delta) \log n} N. \tag{4}$$

For rings $P_{i,j}$ inside $P_{i,j^*}$, i.e. $j < j^*$, we have $|S_{i,j}| < \gamma$. By symmetry, we can prove that with probability at least $1 - n^{-9}$,

$$\forall j < j^*, |P_{i,j}| \leq \frac{4}{5(1-\delta) \log n} N. \tag{5}$$

By the definition of $T_i^{(r)} := \{x \in P^{(r)} : \widetilde{\mathbf{d}}(x, S^{\mathsf{peel}}) \leq (2C_0)^{j^*} \cdot R\}$, $T_i^{(r)}$ contains every point in $P_{i,j}$ ($j \leq j^*$) and some points in $P_{i,j^*+1}$. Conditioning on (4) and (5), we have

$$\mathrm{cost}\left(T_i^{(r)}, c^*\right) = \sum_{x \in T_i^{(r)}} d(x, c^*) \geq (2C_0)^{j^*-1} \left(|P_{i,j}| + |P_{i,j+1} \cap T_i^{(r)}|\right) \geq \frac{\left|T_i^{(r)}\right| \mathrm{diam}(T_i^{(r)})}{8C_0^2 \log n}. \tag{6}$$

By union bound, (6) holds for every $1 \leq i \leq |\mathsf{WC}|, 1 \leq r \leq 10 \log n$ with probability at least $1 - n^{-7}$. We conclude the proof by showing (6) implies

$$\mathrm{cost}(\mathcal{X}, \mathsf{WC}) = \sum_{i=1}^{|\mathsf{WC}|} \sum_{r=1}^{10 \log n} \mathrm{cost}(T_i^{(r)}, c_i) \geq \frac{1}{8C_0^2 \log n} \sum_{i=1}^{|\mathsf{WC}|} \sum_{r=1}^{10 \log n} \left|T_i^{(r)}\right| \mathrm{diam}(T_i^{(r)}).$$

$\square$

*Proof of Lemma C.2.* By Lemma C.4, C.5, C.6, for every $i, j, T \subseteq P_{i,j}$, and for every $\mathcal{C} \subset B(c_i, 60\mathrm{diam}(T)/\varepsilon), \Gamma$, Algorithm 5 outputs set $FC(T)$ such that $|\mathrm{cost}(T, \mathcal{C}, \Gamma) - \mathrm{cost}(FC(T), \mathcal{C}, \Gamma)| \leq \varepsilon' \cdot |T| \cdot \mathrm{diam}(T)$ with probability $1 - n^{-10} \cdot O(n^{-k}(n + k\varepsilon^{-1})^{-k})$. Then by applying union bound to the result of Lemma C.3, with probability at least $1 - n^{-8}$, for every $1 \leq i \leq |\mathsf{WC}|, 1 \leq r \leq 10 \log n$, Algorithm 5 outputs set $FC(T_i^{(r)})$ such that for every $\mathcal{C}, \Gamma$,

$$\left|\mathrm{cost}(FC(T_i^{(r)}), \mathcal{C}, \Gamma) - \mathrm{cost}(T_i^{(r)}, \mathcal{C}, \Gamma)\right| \leq \frac{\varepsilon}{10} \cdot \mathrm{cost}(T_i^{(r)}, \mathcal{C}, \Gamma) + \frac{\varepsilon}{10\alpha_{\mathsf{WC}} C_0^2 \log n} \cdot |T_i^{(r)}| \cdot \mathrm{diam}(T_i^{(r)}). \tag{7}$$

Let $\sigma^*$ be the optimal assignment of $\mathrm{cost}(\mathcal{X}, \mathcal{C}, \Gamma)$. Define $\Gamma_i^{(r)}(c) = \sum_{x \in T_i^{(r)}} \sigma(x, c)$. Summing up for every $i, r$, conditioning on (3) and (7), we have

$$\mathrm{cost}(FC, \mathcal{C}, \Gamma) - \mathrm{cost}(\mathcal{X}, \mathcal{C}, \Gamma) \leq \sum_i \sum_r (\mathrm{cost}(FC(T_i^{(r)}), \mathcal{C}, \Gamma_i^{(r)}) - \mathrm{cost}(T_i^{(r)}, \mathcal{C}, \Gamma_i^{(r)}))$$

$$\leq \sum_{i=1}^{|\mathsf{WC}|} \sum_{r=1}^{10 \log^2 n} \frac{\varepsilon}{10} \mathrm{cost}(T_i^{(r)}, \mathcal{C}, \Gamma_i^{(r)}) + \frac{\varepsilon}{10\alpha_{\mathsf{WC}} C_0^2 \log n} \cdot |T| \cdot \mathrm{diam}(T)$$

$$\leq \frac{\varepsilon}{10} \mathrm{cost}(\mathcal{X}, \mathcal{C}, \Gamma) + \frac{4\varepsilon}{5\alpha_{\mathsf{WC}}} \cdot \mathrm{cost}(\mathcal{X}, \mathsf{WC})$$

$$\leq \frac{\varepsilon}{10} \mathrm{cost}(\mathcal{X}, \mathcal{C}, \Gamma) + \frac{4\varepsilon}{5} \cdot \mathrm{cost}(\mathcal{X})$$

$$\leq \varepsilon \cdot \mathrm{cost}(\mathcal{X}, \mathcal{C}, \Gamma).$$

By applying the same argument to the optimal assignment $\widetilde{\sigma}^*$ of $\mathrm{cost}(FC, \mathcal{C}, \Gamma)$, we can prove $\mathrm{cost}(\mathcal{X}, \mathcal{C}, \Gamma) - \mathrm{cost}(FC, \mathcal{C}, \Gamma) \leq \varepsilon \cdot \mathrm{cost}(\mathcal{X}, \mathcal{C}, \Gamma)$ holds with probability at least $1 - n^{-7}$, which concludes the proof. $\square$

# D   A $(1 + \varepsilon)$-Coreset for general $(k, z)$ Clustering

In this section, we generalize our analysis of $k$-median algorithm in Section 3 to $(k, z)$-clustering for $z = O(1)$ (including $k$-means). We first remind readers of the stated guarantees as follows.

**Theorem 3.** *There exists an algorithm in the weak-strong oracle model that, with high probability, computes a $(k, \varepsilon)$ coreset of size $\widetilde{O}(\frac{k^2}{\varepsilon^3})$ for $(k, z)$-clustering with any $z = \Theta(1)$ using $\widetilde{O}(\frac{k^2}{\varepsilon^3})$ strong oracle point queries (or edge queries), $\widetilde{O}(nk)$ weak oracle queries, and $\widetilde{O}(nk + k^2/\varepsilon^3))$ time.*

The algorithm is essentially the same as Algorithm 1, and the analysis follows from the same idea of heavy-hitter sampling + peeling combined with the algorithm in Chen (2009). However, we need to change the analysis in the same manner of Chen (2009), and we need a separate charging argument for the "skipped" rings in the $(k, z)$-clustering setting.

**Lemma D.1.** *The coreset produced by Algorithm 1 is a $(1 + O(\varepsilon))$-coreset for $(k, z)$-clustering with size $O(k^2 \log^6 n/\varepsilon^3)$ for any constant $z$ with high probability.*

Since we use the same algorithm (Algorithm 1), we only use $O(k^2 \log^6 n/\varepsilon^3)$ strong oracle SO queries, which leads to a $(1 + \varepsilon)$-coreset for $(k, z)$-clustering with $k^2 \operatorname{polylog}(n)/\varepsilon^3$ SO queries. We will rescale $\varepsilon$ by $\operatorname{polylog}(n)$ factors in the end and argue that the size and number of strong oracle queries are at most $k^2 \operatorname{polylog}(n)/\varepsilon^3$ for any constant $z$.

**Additional notation and generalized triangle inequality.** We give some self-contained notation used in the analysis for the $(k, z)$ and the generalized triangle inequality. For the distance between two points $x$ and $y$, we use $\mathbf{d}^z(x, y)$ to denote the $z$-th power of the distance. For any center $c$ in a clustering $\mathcal{C}$, the *cost* of point $x$ assigned to center $c$ could therefore be expressed as $\operatorname{cost}_z(c, x) = \mathbf{d}^z(c, x)$. Similarly, we could use

$$\operatorname{cost}_z(c, P) = \sum_{\substack{x \in P \\ \mathcal{C}(x)=c}} \mathbf{d}^z(c, x)$$

for the cost of a center on a set of points $P$. Correspondingly, we could denote the total cost

$$\operatorname{cost}_z(\mathcal{C}, P) := \sum_{c \in \mathcal{C}} \operatorname{cost}_z(c, P).$$

The following lemma characterizes the *generalized triangle inequality*.

**Proposition 4.** *Let $(u, v, w)$ be points from a metric space as a subset of $[\Delta]^d$. Then, for any $z \geq 1$, we have*

$$\mathbf{d}^z(u, v) \leq 2^z \cdot \left( \mathbf{d}^z(u, w) + \mathbf{d}^z(w, v) \right).$$

The lemma could be straightforwardly proved by taking the $z$-th power of the triangle inequality and using Jensen's inequality. We omit the proof for the sake of conciseness.

**The approximation analysis.** Similar to the analysis of the $k$-median case, the proof for the approximation guarantees is divided into the guarantees for the rings with samples added to SC and the rings without. For the purpose of conciseness, we omit the proof of many lemmas that directly follow in the exactly same way as in Section 3.

**Step I: the convergence of the algorithm and the structural results.** We argue that all the results in this step follow in the same way as in Section 3. In what follows, we list the lemmas and provide brief justifications for why they continue to hold for general $(k, z)$-clustering objectives.

**Lemma B.1.** *Let $j^*$ be the index of the ring found by line 4(c)iv of Algorithm 1. Then, with probability at least $1 - 1/\operatorname{poly}(n)$, no points in $P_{i,\ell}$ for $\ell > j^* + 1$ is removed in the peeling step.*

**Lemma B.2.** *With probability at least $1 - 1/\operatorname{poly}(n)$, Algorithm 4 (the CONSERVATIVE-PEELING algorithm) could only be executed once. Furthermore, after the execution of Algorithm 4, we have that*

- *all points $x$ such that $\mathbf{d}(x, c_i) \leq \frac{R}{2C_0}$ are removed from $\widetilde{P}_i$.*

- *no points in $P_{i,1}$ is removed.*

**Lemma B.3.** *Let $j^*$ be the index of the ring found by line 4(c)iv of Algorithm 1. Then, with probability at least $1 - 1/\operatorname{poly}(n)$, all points in $P_{i,\ell}$ for $\ell \leq j^*$ are removed in the peeling step.*

The proofs of Lemmas B.1 to B.3 are based on the high-probability success of the distance estimation algorithm of Proposition 3. Therefore, switching to the $(k, z)$-clustering objective does *not* affect the proof.

Since Lemmas B.1 to B.3 continue to hold, the convergence of the algorithm is similarly true.

**Lemma B.4.** *With probability at least $1 - 1/\operatorname{poly}(n)$, the following statements for Algorithm 1 are true:*

- *Each ring $P_{i,j}$ is marked as "processed" by exactly one iteration of line 4c.*

- *After $O(\log^2 n)$ iterations of line 4c, all points in $P_i$ is marked as "processed".*

Conditioning on the high-probability events of Lemmas B.1 to B.3, the proof of Lemma B.4 only uses the concentration of sampled points, which is independent of the change of the objectives.

**Step II: handling the rings with many samples.** Similar to the case of $k$-median, we now handle the rings that are marked "*processed*" by Algorithm 2 (line 4(c)v). The analysis similarly follows from (Chen, 2009). In the first step, we directly use the lemma that establishes the estimation of the number of points in the ring.

**Lemma B.6.** *Let $S$ with $s_r = O(k \log^3 n/\varepsilon^3)$ be the set of points sampled uniformly at random from $\widetilde{P}_i$ and let $S_{i,j} = P_{i,j} \cap S$. For all $j$ such that $|S_{i,j}| \geq 30k \log^2 n/\varepsilon^2$, let $S_{i,j}^{\mathsf{est}}$ be any 1/3 fraction of points from $S_{i,j}$, and $\widetilde{m}_{i,j} = \frac{3|\widetilde{P}_i|}{s_r} \cdot |S_{i,j}^{\mathsf{est}}|$. For all $|P_{i,j}| \geq 30 \log n/\varepsilon^2$, with probability at least $1 - \frac{1}{\mathrm{poly}\, n}$ we have $\widetilde{m}_{i,j} \in [(1-\varepsilon)|P_{i,j}|, (1+\varepsilon)|P_{i,j}|]$.*

We now prove the approximation guarantee of the coreset points that are added by Algorithm 2.

**Lemma D.2.** *For a ring $P_{i,\ell}$ that is marked "processed" by Algorithm 2, let $\mathcal{C}$ be any set of centers. Let $S_{i,\ell}$ denote the set of points in $S \cap P_{i,\ell}$ that are uniformly sampled and $S_{i,\ell}^{\mathsf{weight}}$ be any 2/3 points in $S_{i,\ell}$ that are assigned weight $w = \widetilde{m}_{i,\ell}/|S_{i,\ell}^{\mathsf{weight}}|$ and added to $\mathsf{SC}$. Then, with probability at least $1 - \lambda$*

$$\left| \mathrm{cost}(C, P_{i,\ell}) - \mathrm{cost}(C, S_{i,\ell}^{\mathsf{weight}}) \right| \leq 2^{z+1}\varepsilon \cdot |P_{i,\ell}| \cdot [\mathbf{d}^z(\mathcal{C}, P_{i,\ell}) + (\mathrm{diam}(P_{i,\ell}))^z]$$

*Proof.* We follow the same proof strategy as Lemma B.8. For $p \in P_{i,\ell}$, let $h(p) = \mathbf{d}^z(\mathcal{C}, p)$. We have,

$$0 \leq h(p) = \mathbf{d}^z(\mathcal{C}, p) \leq [\mathbf{d}(\mathcal{C}, P_{i,\ell}) + \mathrm{diam}(P_{i,\ell})]^z \leq 2^z [\mathbf{d}^z(\mathcal{C}, P_{i,\ell}) + (\mathrm{diam}(P_{i,\ell}))^z].$$

Let $\eta = 0$, $M = 2^z [\mathbf{d}^z(\mathcal{C}, P_{i,\ell}) + (\mathrm{diam}(P_{i,\ell}))^z]$ and $\delta = \varepsilon M$. For $\left|S_{i,\ell}^{\mathsf{weight}}\right| \geq (M^2/2\delta^2) \ln 2/\lambda$, from Lemma B.7 we have

$$\Pr\left[\left|\frac{\sum_{p \in P_{i,\ell}} \mathbf{d}^z(\mathcal{C}, p)}{|P_{i,\ell}|} - \frac{\sum_{s \in S_{i,\ell}^{\mathsf{weight}}} \mathbf{d}^z(\mathcal{C}, s)}{\left|S_{i,\ell}^{\mathsf{weight}}\right|}\right| \geq \varepsilon \cdot 2^z [\mathbf{d}^z(\mathcal{C}, P_{i,\ell}) + (\mathrm{diam}(P_{i,\ell}))^z]\right] \leq \lambda \tag{8}$$

Using the above equation and conditioning on the high probability event of Lemma B.6, we get

$$\left|\mathrm{cost}_z(\mathcal{C}, P_{i,\ell}) - \mathrm{cost}_z(\mathcal{C}, S_{i,\ell}^{\mathsf{weight}})\right| = \left|\sum_{p \in P_{i,\ell}} \mathbf{d}^z(\mathcal{C}, p) - \sum_{s \in S_{i,\ell}^{\mathsf{weight}}} \mathbf{d}^z(\mathcal{C}, s)w(s)\right|$$

$$= |P_{i,\ell}| \left|\frac{\sum_{p \in P_{i,\ell}} \mathbf{d}^z(\mathcal{C}, p)}{|P_{i,\ell}|} - \frac{\sum_{s \in S_{i,\ell}^{\mathsf{weight}}} \mathbf{d}^z(\mathcal{C}, s)w(s)}{|P_{i,\ell}|}\right|$$

$$\leq |P_{i,\ell}| \left|\frac{\sum_{p \in P_{i,\ell}} \mathbf{d}^z(\mathcal{C}, p)}{|P_{i,\ell}|} - \frac{(1-\varepsilon)\sum_{s \in S_{i,\ell}^{\mathsf{weight}}} \mathbf{d}^z(\mathcal{C}, s)|P_{i,\ell}|}{|P_{i,\ell}|\left|S_{i,\ell}^{\mathsf{weight}}\right|}\right|$$

$$\text{(From Lemma B.6 and } w = \widetilde{m}_{i,\ell}/|S_{i,\ell}^{\mathsf{weight}}|)$$

$$\leq |P_{i,\ell}| \left|\frac{\sum_{p \in P_{i,\ell}} \mathbf{d}^z(\mathcal{C}, p)}{|P_{i,\ell}|} - \frac{\sum_{s \in S_{i,\ell}^{\mathsf{weight}}} \mathbf{d}^z(\mathcal{C}, s)}{\left|S_{i,\ell}^{\mathsf{weight}}\right|}\right| + \varepsilon|P_{i,\ell}|\frac{\sum_{s \in S_{i,\ell}^{\mathsf{weight}}} \mathbf{d}^z(\mathcal{C}, s)}{\left|S_{i,\ell}^{\mathsf{weight}}\right|}$$

$$\leq |P_{i,\ell}|\varepsilon \cdot 2^z [\mathbf{d}^z(\mathcal{C}, P_{i,\ell}) + (\mathrm{diam}(P_{i,\ell}))^z] + \varepsilon|P_{i,\ell}| \cdot 2^z [\mathbf{d}^z(\mathcal{C}, P_{i,\ell}) + (\mathrm{diam}(P_{i,\ell}))^z]$$

$$\leq 2^{z+1}\varepsilon \cdot |P_{i,\ell}| \cdot [\mathbf{d}^z(\mathcal{C}, P_{i,\ell}) + (\mathrm{diam}(P_{i,\ell}))^z]$$

w.p. at least $1 - \lambda$. Second last inequality follows from, for $s \in S_{i,\ell}^{\mathsf{weight}}$, $\mathbf{d}^z(\mathcal{C}, s) \leq 2^z [\mathbf{d}^z(\mathcal{C}, P_{i,\ell}) + (\mathrm{diam}(P_{i,\ell}))^z]$. $\quad\square$

We now bound the costs for the rings with points added to the strong coreset $\mathsf{SC}$ in the same way as in Lemma B.9.

**Lemma D.3.** *Let $\mathcal{P}^{\mathsf{heavy}}$ be the set of rings with points being added to* SC. *Furthermore, let*

$$\mathrm{cost}_z(\mathcal{C}, \mathcal{P}^{\mathsf{heavy}}) := \sum_{P_{i,j} \in \mathcal{P}^{\mathsf{heavy}}} \mathrm{cost}_z(\mathcal{C}, P_{i,j})$$

*be the total cost with respect to any set of center $\mathcal{C} \subseteq X$ of size at most $k$, induced by the rings in $\mathcal{P}^{\mathsf{heavy}}$, and let*

$$\widetilde{\mathrm{cost}}^z(\mathcal{C}, \mathcal{P}^{\mathsf{heavy}}) := \sum_{P_{i,j} \in \mathcal{P}^{\mathsf{heavy}}} \mathrm{cost}_z(\mathcal{C}, S_{i,j}^{\mathsf{weight}}).$$

*Then, with probability at least $1 - 1/\mathrm{poly}\, n$, we have*

$$\left| \mathrm{cost}_z(\mathcal{C}, \mathcal{P}^{\mathsf{heavy}}) - \widetilde{\mathrm{cost}}^z(\mathcal{C}, \mathcal{P}^{\mathsf{heavy}}) \right| \leq 2^{z+1}\varepsilon \cdot \sum_{P_{i,j} \in \mathcal{P}^{\mathsf{heavy}}} |P_{i,j}| \cdot [\mathbf{d}^z(\mathcal{C}, P_{i,\ell}) + (\mathrm{diam}(P_{i,\ell}))^z].$$

*Proof.* Fix some set of centers $\mathcal{C}$ of size at most $k$. Let $\lambda = \Lambda / \left( n^k(ck \log \beta n) \right)$ for some constant $c$. From Lemma D.2 for a particular $i$ and $\ell$, we have with probability at least $1 - \lambda$

$$\left| \mathrm{cost}_z(\mathcal{C}, P_{i,\ell}) - \mathrm{cost}_z(\mathcal{C}, S_{i,\ell}^{\mathsf{weight}}) \right| \leq 2^{z+1}\varepsilon \cdot |P_{i,\ell}| \cdot [\mathbf{d}^z(\mathcal{C}, P_{i,\ell}) + (\mathrm{diam}(P_{i,\ell}))^z]$$

This holds for the points added to SC by Algorithm 2 as the number of points added for each heavy ring is $\left| S_{i,\ell}^{\mathsf{weight}} \right| \geq 30 \cdot k \log^2 n / \varepsilon^2 \geq (1/2\varepsilon^2) \ln 2/\lambda = (1/2\varepsilon^2) \left( \ln(cn^k k \log \beta n) + \ln \frac{1}{\Lambda} \right)$

Now, since we have $k$ centers in the approximate solution and each center has at most $O(\log \beta n)$ rings, union bound over all the rings gives us with probability at least $1 - \Lambda/n^k$,

$$\left| \mathrm{cost}_z(\mathcal{C}, \mathcal{P}^{\mathsf{heavy}}) - \widetilde{\mathrm{cost}}^z(\mathcal{C}, \mathcal{P}^{\mathsf{heavy}}) \right| \leq \sum_{P_{i,\ell} \in \mathcal{P}^{\mathsf{heavy}}} \left| \mathrm{cost}_z(\mathcal{C}, P_{i,\ell}) - \mathrm{cost}_z(\mathcal{C}, S_{i,\ell}^{\mathsf{weight}}) \right|$$

$$\leq 2^{z+1}\varepsilon \cdot \sum_{P_{i,\ell} \in \mathcal{P}^{\mathsf{heavy}}} |P_{i,\ell}| \cdot [\mathbf{d}^z(\mathcal{C}, P_{i,\ell}) + (\mathrm{diam}(P_{i,\ell}))^z]$$

As there are at most $n^k$ choices of $k$ centers $\mathcal{C}$ from point set $X$ of size $n$, the proof concludes with a final union bound over the choice of centers $\mathcal{C}$ and setting $\Lambda = 1/n^{c'}$. $\qquad\square$

**Step III: handling the rings with few samples.** We now analyze the rings that are marked "*processed*" without any points added to SC in Algorithm 1. We need to define a new charging scheme that works with $\mathrm{diam}^z(P_{i,\ell})$ and bound the loss by generalized triangle inequality. We first note that the size relationship between the ring $P_{i,\ell}$ and the ring $P_{i,j^*}$ that marks $P_{i,\ell}$ as "*processed*" continues to hold.

**Lemma B.10.** *Let ring $P_{i,\ell}$ be marked as "*processed*" in iteration $r$ without any point $x \in P_{i,\ell}$ being added to the coreset* SC. *Furthermore, let $j^*$ be the ring found by line 4(c)iv of iteration $r$. Then, with probability at least $1 - 1/\mathrm{poly}(n)$, we have*

$$|P_{i,j^*}| \geq \frac{2C_0}{\varepsilon} \cdot |P_{i,\ell}|.$$

The proof of the lemma only requires the heavy-hitter sampling argument which is independent of the objectives. We now define a generalized *charging scheme* similar to that of Algorithm 6, albeit working with $\mathrm{diam}^2(P_{i,\ell})$. For each ring $P_{i,\ell}$ marked as "*processed*" without points being added to SC, we define $j^*(\ell)$ as be the index of ring $P_{i,j^*}$ that marked $P_{i,\ell}$ as "*processed*".

---

**Algorithm 7.  A charging scheme** (a thought process for the **analysis purpose** only).

- For each ring $P_{i,\ell}$ such that $\ell$ is marked as "*processed*" in iteration $r$ without any point $x \in P_{i,\ell}$ being added to the coreset $\mathsf{SC}$:

    - If $j^*(\ell) \neq 0$, we write a charge of $\frac{\varepsilon}{2C_0} \cdot \mathrm{diam}^z(P_{i,\ell})$ to *all* points in $P_{i,j^*(\ell)}$.
    - Otherwise, if $j^*(\ell) = 0$ (which implies $\ell = 1$), we write a charge of $\frac{\varepsilon}{2C_0} \cdot \mathrm{diam}^z(P_{i,1})$ to *all* points in $\{x \in P_{i,0} \mid \mathbf{d}(x, c_i) \geq \frac{R}{2C_0}\}$.

- If ring $P_{i,\ell}$ receives charges of at most $\gamma$ on each point *and* $\ell$ is marked as "*processed*" without any point $x \in P_{i,\ell}$ being added to the coreset $\mathsf{SC}$, then *transfer* the charges by writing charges of $\frac{\varepsilon}{2C_0} \cdot \gamma$ to *all* points in $P_{i,j^*(\ell)}$.

- Continue recursively until all charges are written on points of rings $P_{i,\ell}$ such that points $x \in P_{i,\ell}$ are added to the coreset $\mathsf{SC}$.

---

We now give two lemmas that characterize the guarantees of the charging scheme in the same way as the $k$-median case.

**Lemma D.4.** *For any ring $P_{i,\ell}$, conditioning on the high-probability event of Lemma B.10, the total number of charges Algorithm 6 could distribute is at least $|P_{i,\ell}| \cdot \mathrm{diam}^z(P_{i,\ell})$.*

*Proof.* The lemma follows from the proof of Lemma D.4 with the change of the $z$-th power. For each ring $P_{i,\ell}$, we claim that conditioning on the high-probability event of Lemma B.10, there is $\left| P_{i,j^*(\ell)} \right| \geq \frac{C_0}{50\varepsilon} \cdot |P_{i,\ell}|$. The statement trivially holds if $j^*(\ell) \neq 0$. On the other hand, if $j^*(\ell) = 0$, by the rules in Algorithm 1 (which implies $\ell = 1$), we argue that there are $\frac{C_0}{50\varepsilon} \cdot |P_{i,1}|$. In particular, we could treat $P_{i,0} \setminus P_{i,0}^{\mathrm{close}}$ (defined in Algorithm 7) as a separate ring, and apply Lemma B.10 to get the desired result. Therefore, the total charge $P_{i,\ell}$ could write to is at least

$$\frac{\varepsilon}{2C_0} \cdot \mathrm{diam}^z(P_{i,\ell}) \cdot \left| P_{i,j^*(\ell)} \right| \geq \mathrm{diam}^z(P_{i,\ell}) \cdot |P_{i,\ell}|,$$

as claimed. $\square$

Next, we again bound the number of charges for any point in a ring that could be received.

**Lemma D.5.** *For any ring $P_{i,\ell}$, conditioning on the high-probability event of Lemma B.10, for any point $x \in P_{i,\ell}$, the total number of charges written on $x$ by Algorithm 6 is at most $\varepsilon \cdot (2C_0)^z \cdot \mathrm{diam}^z(P_{i,\ell+1})$ for any $\varepsilon < 1$.*

*Proof.* We refer the reader to Lemma B.12 for the full proof, and we only give a concise version here that skips some of the intermediate reasoning steps. Assuming no recursive charging happens, the total charge for $P_{i,j^*}$ could receive is at most

$$\sum_q \mathsf{charge}(j^* \leftarrow q) \leq \sum_{q=0}^{j^*+1} \frac{\varepsilon}{2C_0} \cdot \mathrm{diam}^z(P_{i,q})$$

$$\leq \frac{\varepsilon}{2C_0} \cdot \mathrm{diam}^z(P_{i,j^*+1}) \cdot \sum_{q=0}^{j^*+1} \cdot (\frac{1}{2C_0})^{j^*-q+1}$$

$$\leq \frac{\varepsilon}{C_0} \cdot \mathrm{diam}^z(P_{i,j^*+1}). \qquad (\sum_{q=0}^{j^*+1} \cdot (\frac{1}{2C_0})^{j^*-q+1} \leq 2 \text{ for any } j^* \geq 0)$$

On the other hand, if recursive charging happens, with the same argument as in Lemma B.12, we have that

$$\frac{\varepsilon}{2C_0} \cdot \frac{\varepsilon}{C_0} \cdot \mathrm{diam}^z(P_{i,\ell+1}) \leq \varepsilon^2 \cdot \frac{1}{2C_0^2} \cdot (2C_0)^z \cdot \mathrm{diam}(P_{i,\ell}) \leq (2C_0)^z \cdot \varepsilon^2 \cdot \mathrm{diam}(P_{i,\ell}).$$

Hence, the total amount of charges that could be transferred to any such $j^*$ is at most

$$\sum_{\ell=j^*+1}^{\log n} (2C_0)^z \cdot \varepsilon^{2 \cdot (j^*-\ell)} \cdot \mathrm{diam}(P_{i,j^*}) \leq \varepsilon (2C_0)^z \cdot \cdot \mathrm{diam}(P_{i,j^*+1}),$$

which is as desired. $\square$

We could now bound the total cost contribution from the rings that are marked as "*processed*" without points added to the coreset SC as follows.

**Lemma D.6.** *Let $P_{i,\ell}$ be a ring such without points added to* SC*, and $P_{i,j^*(\ell)}$ be the ring for $P_{i,\ell}$ to charge to in the charging scheme of Algorithm 6. Then, we have that*

$$|P_{i,\ell}| \cdot \operatorname{diam}^z(P_{i,\ell}) \le \varepsilon \cdot (2C_0)^{2z} \cdot \left|P_{i,j^*(\ell)}\right| \cdot \operatorname{diam}^z(P_{i,j^*(\ell)}).$$

*Proof.* The proof is similar to the proof of Lemma D.6. By Lemma D.5, any point in ring $P_{i,j^*(\ell)}$ receives at most $\varepsilon \cdot (2C_0)^z \cdot \operatorname{diam}(P_{i,j^*(\ell)+1})$ charges. We now claim that $\operatorname{diam}(P_{i,j^*(\ell)+1}) \le (2C_0)^z \cdot \operatorname{diam}(P_{i,j^*(\ell)})$. For all rings *except* $j^*(\ell) = 0$, the statement simply follows directly from the construction. If $j^*(\ell) = 0$, by the rule of Algorithm 1, the charges could only be written on $P_{i,0} \setminus P_{i,0}^{\text{close}}$ by the charging rule. Therefore, we have $\operatorname{diam}(P_{i,j^*(\ell)+1}) \le (2C_0)^z \cdot \operatorname{diam}(P_{i,j^*(\ell)})$. Furthermore, we could show that the number of points receiving charges in ring $P_{i,j^*(\ell)}$ is at least the number of points in $P_{i,\ell}$. Therefore, we have that

$$|P_{i,\ell}| \cdot \operatorname{diam}^z(P_{i,\ell}) \le \varepsilon \cdot (2C_0)^z \cdot \left|P_{i,j^*(\ell)}\right| \cdot \operatorname{diam}^z(P_{i,j^*(\ell)+1}) \le \varepsilon \cdot (2C_0)^{2z} \cdot \left|P_{i,j^*(\ell)}\right| \cdot \operatorname{diam}^z(P_{i,j^*(\ell)}),$$

as desired by the lemma statement. □

The next lemma is an analog of Lemma B.14 that bounds the costs between $(\mathcal{C}, P_{i,j})$ and $(\mathcal{C}, P_{i,\star})$ such that $P_{i,\star}$ marked $P_{i,j}$ as "*processed*".

**Lemma D.7.** *Let $\mathcal{C}$ be any set of centers. Then, for any ring $P_{i,j^*}$ and rings $P_{i,j}$ such that $j \le j^* + 1$ and $|P_{i,j}| \le \eta \cdot |P_{i,j^*}|$, we have that*

$$\operatorname{cost}_z(\mathcal{C}, P_{i,j}) \le 2^z \cdot \eta \cdot \operatorname{cost}_z(\mathcal{C}, P_{i,j^*}) + (2C_0)^{2z} \cdot |P_{i,j}| \cdot \operatorname{diam}^z(P_{i,j}).$$

*Proof.* We use generalized triangle inequality (Proposition 4) to prove the lemma. Let $x \in P_{i,j}$ be a point in the ring $P_{i,j}$, we could find a corresponding a point $y \in P_{i,j^*}$ assigned to center $c \in \mathcal{C}$ such that

$$\begin{aligned}
\operatorname{cost}_z(\mathcal{C}, x) &\le \mathbf{d}^z(c, x) &&(\operatorname{cost}_z(\mathcal{C}, x) \text{ is the optimal cost for } x)\\
&\le 2^z \cdot (\mathbf{d}^z(c, y) + \mathbf{d}^z(x, y)) &&(\text{by Proposition 4})\\
&\le 2^z \cdot \operatorname{cost}_z(\mathcal{C}, y) + 2^z \cdot \operatorname{diam}^z(P_{i,j^*})\\
&\le 2^z \cdot \operatorname{cost}_z(\mathcal{C}, y) + 2^z \cdot (2C_0)^z \cdot \operatorname{diam}^z(P_{i,j}). &&(\text{by our construction of the rings and } j \le j^* + 1)
\end{aligned}$$

Iterating the argument with the least-cost point $y \in P_{i,j^*}$ for $|P_{i,j}| \le \eta \cdot |P_{i,j^*}|$ time gives us

$$\begin{aligned}
\operatorname{cost}_z(\mathcal{C}, P_{i,j}) &= \sum_{x \in P_{i,j}} \operatorname{cost}_z(\mathcal{C}, x)\\
&\le 2^z \cdot \sum_{x \in P_{i,j}} \operatorname{cost}_z(\mathcal{C}, y) + 2^z \cdot (2C_0)^z \cdot \operatorname{diam}(P_{i,j})\\
&\le 2^z \cdot \eta \cdot \operatorname{cost}_z(\mathcal{C}, P_{i,j^*}) + (2C_0)^{2z} \cdot |P_{i,j}| \cdot \operatorname{diam}^z(P_{i,j}),
\end{aligned}$$

as desired. □

We can now bound the total cost induced by the "light" rings in the same way of Lemma B.15.

**Lemma D.8.** *Let $\mathcal{P}^{\text{light}}$ be the set of rings that are marked "processed" without points being added to* SC*, and let $\mathcal{P}^{\text{heavy}}$ be all other rings. For any set of centers $\mathcal{C}$, we have that*

$$\operatorname{cost}_z(\mathcal{C}, \mathcal{P}^{\text{light}}) := \sum_{P_{i,j} \in \mathcal{P}^{\text{light}}} \operatorname{cost}_z(\mathcal{C}, P_{i,j}) \le \varepsilon \cdot (2C_0)^{2z} \cdot \log n \cdot \sum_{P_{i,j} \in \mathcal{P}^{\text{heavy}}} \left(\operatorname{cost}_z(\mathcal{C}, P_{i,j}) + |P_{i,j}| \cdot \operatorname{diam}^z(P_{i,j})\right).$$

*Proof.* Similar to the proof of Lemma B.15, we fix any ring $P_{i,j^*} \in \mathcal{P}^{\text{heavy}}$, we let the rings $\mathcal{P}^{\text{light}}(j^*)$ be the set of rings with index $\ell$ such that $j^*(\ell) = j^*$, and apply Lemma D.7 and Lemma D.6 to all rings in $\mathcal{P}^{\text{light}}(j^*)$ and obtain that

$$\sum_{P_{i,j} \in \mathcal{P}^{\text{light}}(j^*)} \operatorname{cost}_z(\mathcal{C}, P_{i,j}) \le \sum_{P_{i,j} \in \mathcal{P}^{\text{light}}(j^*)} \varepsilon \cdot 2^z \cdot \operatorname{cost}_z(\mathcal{C}, P_{i,j^*}) + (2C_0)^{2z} \cdot \sum_{P_{i,j} \in \mathcal{P}^{\text{light}}(j^*)} |P_{i,j}| \cdot \operatorname{diam}^z(P_{i,j})$$

$$(\text{by Lemma D.7})$$

$$\leq \sum_{P_{i,j}\in\mathcal{P}^{\text{light}}(j^*)} \varepsilon\cdot 2^z\cdot\text{cost}_z(\mathcal{C},P_{i,j^*}) + \varepsilon\cdot(2C_0)^{4z}\cdot\sum_{P_{i,j}\in\mathcal{P}^{\text{light}}(j^*)}|P_{i,j^*}|\cdot\text{diam}(P_{i,j^*})$$

(by Lemma B.13)

$$\leq \log n\cdot\left(\varepsilon\cdot 2^z\cdot\text{cost}_z(\mathcal{C},P_{i,j^*}) + \varepsilon\cdot(2C_0)^{4z}\cdot|P_{i,j^*}|\cdot\text{diam}(P_{i,j^*})\right)$$

(at most $\log n$ such rings)

$$\leq \varepsilon\cdot(2C_0)^{4z}\cdot\log n\cdot\left(\text{cost}(\mathcal{C},P_{i,j^*}) + |P_{i,j^*}|\cdot\text{diam}(P_{i,j^*})\right).$$

Once again, since each ring $P_{i,\ell}$ in $\mathcal{P}^{\text{light}}$ has a unique $j^*(\ell)$, summing over all rings in $\mathcal{P}^{\text{light}}$ gives the desired lemma statement. $\qquad\square$

**Wrapping up the proof of Theorem 3.** We now finalize the proof of Lemma D.1. For $(\alpha,\beta)$-approximation for $(k,z)$-clustering, the approximation lower bound for opt is as follows.

**Claim D.9.** *Let* $\text{OPT}^z$ *be the optimal cost of clustering on* $P$ *with* $k$ *centers for* $(k,z)$ *clustering and let* $\mathcal{A}$ *be the centers from a* $\beta$-*approximation solution. Then,* $\sum_{i,j}|P_{i,j}|\left((2C_0)^jR\right)^z \leq (2C_0)^{2z+1}\beta\cdot\text{OPT}^z$ *and* $\sum_{i,j}|P_{i,j}|\text{diam}^z(P_{i,j}) \leq (2C_0)^{3z+1}\beta\cdot\text{OPT}^z$.

*Proof.* Consider any point $p\in P_{i,j}$. For $j=0$, $(2C_0)^jR = R$ and for $j\geq 1$, $(2C_0)^jR \leq 2C_0\mathbf{d}(\mathcal{A},p)$. Hence, for any $j$ we have $\left((2C_0)^jR\right)^z \leq (2C_0\mathbf{d}(\mathcal{A},p)+R)^z \leq 2^z\left((2C_0)^z\mathbf{d}^z(\mathcal{A},p)+R^z\right)$. Now,

$$\sum_{i,j}|P_{i,j}|\left((2C_0)^jR\right)^z = \sum_{i,j}\sum_{p\in P_{i,j}}\left((2C_0)^jR\right)^z \leq \sum_{i,j}\sum_{p\in P_{i,j}}2^z\left((2C_0)^z\mathbf{d}^z(\mathcal{A},p)+R^z\right)$$

$$= 2^z\sum_{p\in X}\left((2C_0)^z\mathbf{d}^z(\mathcal{A},p)+R^z\right)$$

$$\leq 2^z\left((2C_0)^z\beta\cdot\text{OPT}^z + nR^z\right) \leq (2C_0)^{2z+1}\beta\cdot\text{OPT}^z$$

Since $\text{diam}(P_{i,j}) \leq 2(2C_0)^jR$, we have $\sum_{i,j}|P_{i,j}|\left(\text{diam}(P_{i,j})\right)^z \leq (2C_0)^{3z+1}\beta\cdot\text{OPT}^z$ $\qquad\square$

The following lemma directly establishes the desired guarantees as in Theorem 3.

**Lemma D.10.** *For any set of centers* $\mathcal{C}\subseteq X$ *of size at most* $k$, *it holds that*

$$\left|\text{cost}_z(\mathcal{C},X) - \text{cost}_z(\mathcal{C},\mathsf{SC})\right| \leq \varepsilon\cdot 30\beta\cdot(2C_0)^{4z}\cdot\log n\cdot\text{cost}_z(\mathcal{C},X)$$

*with probability at least* $1 - 1/\text{poly}(n)$.

*Proof.* We define $\mathcal{P}^{\text{light}}$ as the rings being marked as "*processed*" without points added to $\mathsf{SC}$ and $\mathcal{P}^{\text{heavy}}$ as the rings with points added to $\mathsf{SC}$. We have

$$\text{cost}_z(\mathcal{C},X) = \sum_{P_{i,j}^{\text{light}}\in\mathcal{P}^{\text{light}}}\text{cost}_z(\mathcal{C},P_{i,j}^{\text{light}}) + \sum_{P_{i,j}^{\text{heavy}}\in\mathcal{P}^{\text{light}}}\text{cost}_z(\mathcal{C},P_{i,j}^{\text{heavy}}).$$

Define $\widetilde{\text{cost}}(\mathcal{C},\mathcal{P}^{\text{light}})$ and $\widetilde{\text{cost}}(\mathcal{C},\mathcal{P}^{\text{heavy}})$ as the cost induced by $\mathsf{SC}$ for $\mathcal{C}$, and it follows that $\text{cost}(\mathcal{C},\mathsf{SC}) = \widetilde{\text{cost}}(\mathcal{C},\mathcal{P}^{\text{light}}) + \widetilde{\text{cost}}(\mathcal{C},\mathcal{P}^{\text{heavy}})$. Furthermore, by our construction, there is $\widetilde{\text{cost}}(\mathcal{C},\mathcal{P}^{\text{light}}) = 0$. Conditioning on the high-probability events of Lemma D.3 and Lemma D.8, we have that

$$\left|\text{cost}(\mathcal{C},\mathcal{P}^{\text{heavy}}) - \widetilde{\text{cost}}(\mathcal{C},\mathcal{P}^{\text{heavy}})\right| \leq \varepsilon\cdot 2^{z+1}\cdot\sum_{P_{i,j}\in\mathcal{P}^{\text{heavy}}}|P_{i,j}|\cdot\left(\mathbf{d}^z(\mathcal{C},P_{i,\ell}) + \text{diam}^z(P_{i,\ell})\right)$$

$$\left|\text{cost}(\mathcal{C},\mathcal{P}^{\text{light}}) - \widetilde{\text{cost}}(\mathcal{C},\mathcal{P}^{\text{light}})\right| \leq \text{cost}_z(\mathcal{C},\mathcal{P}^{\text{light}})$$

$$\leq \varepsilon\cdot(2C_0)^{2z}\cdot\log n\cdot\sum_{P_{i,j}\in\mathcal{P}^{\text{heavy}}}\left(\text{cost}_z(\mathcal{C},P_{i,j}) + |P_{i,j}|\cdot\text{diam}^z(P_{i,j})\right).$$

As such, we could bound the difference of the cost as

$$
\begin{aligned}
&\left|\text{cost}_z(\mathcal{C}, X) - \text{cost}_z(\mathcal{C}, \mathsf{SC})\right| \\
&\leq \left|\text{cost}(\mathcal{C}, \mathcal{P}^{\text{heavy}}) - \widetilde{\text{cost}}(\mathcal{C}, \mathcal{P}^{\text{heavy}})\right| + \left|\text{cost}(\mathcal{C}, \mathcal{P}^{\text{light}}) - \widetilde{\text{cost}}(\mathcal{C}, \mathcal{P}^{\text{light}})\right| \\
&\leq \varepsilon \cdot (2C_0)^{2z} \cdot \log n \cdot \sum_{P_{i,j} \in \mathcal{P}^{\text{heavy}}} |P_{i,j}| \cdot (\mathbf{d}^z(\mathcal{C}, P_{i,\ell}) + \text{diam}^z(P_{i,\ell})) + \varepsilon \cdot (2C_0)^{2z} \cdot \log n \cdot \sum_{P_{i,j} \in \mathcal{P}^{\text{heavy}}} \text{cost}(\mathcal{C}, P_{i,j})
\end{aligned}
$$

$$\text{(by Lemma D.8)}$$

$$
\leq \varepsilon \cdot (2C_0)^{5z+1} \cdot \log n \cdot \text{cost}_z(\mathcal{C}, X) + \varepsilon \cdot (2C_0)^{5z+1} \cdot \log n \cdot \beta \cdot \text{OPT}^z \qquad \text{(by Claim D.9)}
$$

$$
\leq 2\varepsilon \cdot (2C_0)^{5z+1} \cdot \log n \cdot \beta \cdot \text{cost}_z(\mathcal{C}, X),
$$

which is as desired. $\qquad\square$

Finally, we could rescale $\varepsilon$ using $\varepsilon = \frac{\varepsilon'}{2 \cdot (2C_0)^{5z+1} \cdot \log n \cdot \beta} = O(\frac{\varepsilon'}{\log n})$ by the constant choices of $\beta$, $C_0$, and $z$. The size of the coreset and number of $\mathsf{SO}$ queries are therefore $O(k^2 \log^5 n / \varepsilon'^3) = O(k^2 \log^8 n / \varepsilon^3) = k \operatorname{polylog}(n) / \varepsilon^3$, as desired by Theorem 3.