# OpenReview forum: "Relative Error Fair Clustering in the Weak-Strong Oracle Model"
_ICML.cc/2025/Conference — ICML 2025 poster_

### Official Review · Reviewer_6eJr · 2025-03-13

**Overall Recommendation:** 3

**Summary:**

*Background*: This work studies the problem of fair clustering in the weak-strong oracle framework. In this setup, there exists a strong oracle offering precise distance measurements at a higher cost, alongside a weak providing less accurate distance estimates at a lower cost. The goal is to minimize strong oracle queries while achieving a near-optimal fair clustering. Fair clustering refers to addressing faireness concerns in the clustering process – for instance, there are $\ell$ groups in total, and for each group $j$, set fairness constraints to be parameters  $0\leq \alpha_j\leq \beta_j\leq 1$, such that each cluster should have at least $\alpha_j$ and at most $\beta_j$ fraction of points from group $j$.

*Main results*: First, when the number of disjoint groups each point belongs to is bounded by $\Lambda$, the authors designed an algorithm that constructs a $(k,\epsilon)$ fair coreset of size $\tilde{O}(\Lambda k^2/\epsilon^2)$ in the $k$-median clustering with $\tilde{O}(\Lambda k)$ strong oracle point queries. This also implies a corresponding corest for the $\Lambda =1$ case, i.e., the assignment-preserving $k$-median clustering. Then for the $(k,z)$-clustering without fairness constraint and $z=O(1)$, a $(k,\epsilon)$-coreset of size $\tilde{O}(k^2/\epsilon^3)$ is constructed with $\tilde{O}(k^2/\epsilon^3)$ strong oracle point queries.

*Main methodology*: The approach employs a ring sampling technique, improved by heavy-hitter sampling and recursive peeling. During each iteration, the algorithm samples some points, and use strong oracle queries on each sampled points to determine their ring assignments. Then, the rings with more sampled points are denoted as heavy ones, which contribute more to the final solution, so we can build coresets only for these heavy rings. At the end of each iteration, the algorithm peels off the points in the heavy rings. After all the iterations, the union of all coresets constitutes the final fair $k$-clustering. Moreover, to allow assignment-preserving properties hold in the clustering, the algorithm performs an additional sampling step at the end of each iteration.

**Claims And Evidence:**

The claims are supported by rigorous proofs, though some parts are somewhat difficult to follow (see comments below).

**Essential References Not Discussed:**

I did not identify any essential references that are missing from the discussion.

**Experimental Designs Or Analyses:**

I reviewed Experiments section in the appendix. The authors conducted experiments on only two datasets, which may not be sufficient to show the advantages of their approach.

Besides, the chosen baseline might be too simple. It could be worthwhile to consider a baseline using coresets generated by Algorithm 1 without having fairness constraint. This comparison could provide a clearer understanding of the impact of including fairness constraints.

**Methods And Evaluation Criteria:**

The authors computed the fair $k$-median cost for specified centroids and fairlet decomposition, and compared fair $k$-median cost of their proposed algorithm against uniform coresets. This evaluation criteria appears suitable.

**Other Comments Or Suggestions:**

Typos:

- Line 145, left col., change “belong” to “belongs”
- Line 121, right col., delete “such that each $x\in X$”
- Line 123, right col. correct "$\log^4 \cdot \log n$"
- Line 140, right col. it is mentioned that "... with even better efficiency on the number of strong oracle queries". However, in line 154, right col. it is stated that "the number of strong oracle queries is slightly worse". These two statements seem contradicting.
- Line 256, right col., delete the second $s_3$ in “$s_3$ weak oracle queries $s_3$”
- Line 333, right col., change “A updated set” to “An updated set”
- Line 353, right col., change “A updated set” to “An updated set”
- Line 372, right col., change “A updated set” to “An updated set”
- Line 397, left col., the repeated “for” in “for each center for $O(\log^2⁡n)$ iterations” sounds a little unclear. Consider rephrasing it to “for $O(\log^2⁡n)$ queries per center”, or “for each center, we have $O(\log^2⁡n)$ iterations, and in each iteration, …”
- Line 766~767, the sentence “Our plan is to show that in each iteration of line 4c, there are i). each iteration of …; ii). for any ring …; and iii). for any …” has grammatical inconsistencies. Consider rephrasing to: “Our plan is to show that i). in each iteration of …; ii). for any ring …; iii). for any…”
- Line 354, left col., change $S_{i,\ell}$ to $S_{i,j^*}$
- Line 1550, there is a missing “)” of $diam(T)$
- Line 1306, should it be $Pr⁡[\cdots\geq \cdots]≤\delta$? (According to the reference on Line 1550~1551)

Suggestions:
- it will be helpful to discuss the state-of-the-art of the sizes of coreset for $(k,z)$-clustering, in particular, for $k$-median

**Other Strengths And Weaknesses:**

Strengths:

- the first $(1+\epsilon)$-coresets for fair $k$-median clustering using $poly((k\log n)/\epsilon)$ queries to the strong oracle.

Weaknesses:

- the experiments are somewhat too simplistic (see comments above).

- The proofs and inequalities are hard to follow and verify, and need clearer explanations. It would be better if inequalities have tags or comments to explain on them.

**Questions For Authors:**

In Theorem 4, the coreset for $(k, z)$-clustering has a size of $\tilde{O}(k^2/\epsilon^3)$ with $\tilde{O}(k^2/\epsilon^3)$ strong oracle queries. In particular, this (Theorem 15) applies to $k$-median clustering. However, in Theorem 3, even with the fairness constraint, you construct a coreset for $k$-median with smaller size $\tilde{O}(k^2/\epsilon^2)$ and with a significantly smaller $\tilde{O}(k)$ strong oracle query complexity.

I understand that the algorithm for Theorem 4 likely applies to a more general problem. However, the discrepancy mentioned above is still somewhat counterintuitive. Could you clarify this?

**Relation To Broader Scientific Literature:**

The weak-strong oracle model is motivated by the fact that modern machine learning models often use embedding functions to estimate distances for non-metric data, which can be computationally expensive, and by the applications that involve trade-offs between information accuracy and price. The concept of fair clustering arises from the need of incorporating fairness into clustering. This paper contributes to algorithms in fair clustering within weak-strong oracle model, which is a potential application direction in the filed of modern machine learning.

**Theoretical Claims:**

I performed a sanity check of the proofs for the following: Lemma 3.1, Lemma B.1, Lemma B.2, Lemma B.3, Claim B.5, Lemma B.4, Lemma B.6, Lemma B.7 (B.7 is a result from another paper). The following are some issues that need to be addressed:

- Lemma 3.1: This lemma states that “the algorithm converges in xxx time”. If this refers to the running time of the algorithm, it should account not only for the weak-strong model queries but also for the running time of other invoked subroutines. For example, in Line v of Algorithm 1, there is a “for” loop over $\ell \leq j^*+1$, and the algorithm invokes the Coreset-Update subroutine over each $\ell$. Should the running time include invocations? While the running time might remain unchanged after considering these subroutine calls, it could be beneficial to include this explanation in the proof for clarity.

- Lemma B.2: Line 811~812, the inequality $\tilde{d}(y,S^{pell})\geq \tilde{d}(y,c_i)\geq R$ is not explained.

- Lemma C.2 (Page 29):

-- The authors didn’t explain on $\alpha_{WC}$; it may refer to the approximation ratio of $cost(X,WC)$ regarding optimal cost $cost(X)$, and thus $\alpha_{WC}>1$. From Line 1560 to Line 1563, it follows from Equation (7), but it seems that $\alpha_{WC}<1$; while from Line 1566 to Line 1568, it seems that $\alpha_{WC}>1$. These two derivations look like a contradiction on the range of $\alpha_{WC}$.

-- The first inequality of $cost(FC,C,\Gamma)-cost(X,C,\Gamma)$ is not explained.

-- Besides, this derivation is not explained: $\sum\sum cost(T_i^{(r)},C,Γ_i^{(r)})≤cost(X,C,\Gamma)$.

- Lemma C.4: Line 1404, need to explain $p$ in the $\sigma(p,c)$. Perhaps it should be $x$, instead of $p$?

---

> ### Author Rebuttal · Authors · 2025-03-31
>
> We thank the reviewer for their comments and address their questions below.
>
>  ### Re: Runtime claim in Lemma 3.1
> The bound is indeed accounting for the runtime of subroutines. To see this note that for ``not processed’’ rings we check the size in the sampled set and then perform peeling. If the sample size is large enough, we add it to coreset or ignore otherwise. This takes $\tilde{O}(k^2/\varepsilon^3)$ time for $O(\log n)$ rings for $k$ centers. For the peeling subroutine, we perform $O(\log n)$ WO queries for at most $n$ points and $k$ centers to estimate the distances, giving the final runtime of $\tilde{O}(nk + k^2/\varepsilon^3)$. We will expand to make this clearer.
>
>
> ### Re: Lemma B.2
> The term $\tilde{d}(y, c_i)$ should be $d(y, c_i)$, thank you for catching the typo. Following this, notice that $c_i$ is part of the set $S^{\text{peel}}$ by definition and hence the inequality follows from Proposition 13 and definition of rings.
>
>
> ### Re: Lemma C.2
> \alpha_{WC}: Yes, \alpha_{WC} is the cost of the weak coreset (defined on line 1300~1301). There is a typo in equation (7), which should be eps / (10 alpha_WC C_0^2 log n) instead of eps / (10 C_0^2 log n). So we just need \alpha_{WC} > 1.
>
> The first inequality of cost(FC, C, \Gamma) - cost(X, C, \Gamma):
> There is a typo in the RHS of line 1560~1561, should be cost(FC(T_i^{(r)}),C,\Gamma^{(r)}_i)) - cost(T_i^{(r)},C,\Gamma^{(r)}_i) (i.e. swap the two terms). Here, we are partitioning T and FC into rings. Summing up the optimal cost of these rings should get a total cost no less than the original cost of the whole set (before partition). This is because we fix the assignment constraint (i.e. \Gamma^{(r)}_i) during the partition process. So we have cost(FC, C, \Gamma) <= \sum_i \sum_r cost(FC(T_i^{(r)}),C,\Gamma^{(r)}_i).
>
> Since sigma^* is the optimal assignment of cost(X, C, \Gamma), letting the assignment constraint be the optimal one does not increase the cost, so we have cost(X, C, \Gamma) = \sum_i \sum_r cost(T_i^{(r)},C,\Gamma^{(r)}_i).
>
>
> ### Re: Lemma C.4
> Yes, it should be $\sigma(x,c)$, we will change it.
>
>
>
> ### Re: Experiments
> The choice of our baseline of uniformly sampled coreset stems from comparing our method to another that uses a comparable number of SO queries. We do not compare it to unconstrained k-median as fairness constraints typically only increase the cost of clustering compared to optimal unconstrained one.
>
>
> ### Re: Coreset sizes in Thm 3 and Thm 4
> The reviewer’s intuition is correct that the value of Theorem 4 is that it applies to general $(k,z)$-clustering for any $z=O(1)$ (instead of only $k$-median). Due to the page limits of ICML, we decided to present the $k$-median version for the $(1+\varepsilon)$ coreset (without fairness) in the main paper to help understanding. The actual proof of the more general Theorem 4 is in Appendix D: if follows the same algorithm as in Theorem 15, and the analysis is different. At the moment, it is unclear to us how to generalize the analysis of Theorem 3 to the general $(k,z)$-setting. We will clarify this in future versions.
>
>
> ### Re: Typos and suggestions
> We thank the reviewer for the careful reading and their suggestions. We will make the necessary corrections and changes.

---

### Official Review · Reviewer_mTF8 · 2025-03-13

**Overall Recommendation:** 4

**Summary:**

The authors study the fair $(k,z)$-clustering problem in a weak-strong oracle model. Each data point may belong to one or more groups, and the goal is to cluster the data points while minimizing a given clustering objective. The fairness requirement ensures that within each cluster, data points from different groups are represented within a specified range, i.e., $\alpha_i \leq \beta_i$ for each group $i$.

In this setting, querying the strong oracle, which computes the exact distances between data points, is expensive. Instead, a weaker oracle is available, which provides accurate distance predictions for at least $\frac{2}{3}$ of the data points while returning arbitrary values for the remaining portion. Since it is unknown which data points have inaccurate predictions from the weak oracle, repeating queries to improve accuracy is not applicable. The goal is to minimize the number of queries to the strong oracle and obtain a solution that optimizes the clustering objective.

The high-level idea of the paper is to construct a coreset of small size while minimizing the number of queries to the strong oracle. If the coreset is sufficiently small, the strong oracle can be used with a quadratic number of queries to compute exact distances for the data points in the coreset. Since the coreset is small, existing clustering methods can be applied efficiently to obtain the final clustering solution.

**[Update after rebuttal]** I am satisfied with the responses of authors and, I will retain my assessment unchanged.

**Claims And Evidence:**

Though the paper is theory-heavy, the authors do a commendable job of explaining the high-level idea and providing a clear overview of their approach. Additionally, before and after each lemma, the high-level explanations are sufficiently clear to convey the main intuition, assuming one trusts the authors’ claims about the detailed proofs in the appendix.

I have reviewed the main text and parts of the appendix, but given the review timeline, I was unable to verify the proofs in detail. Since most of the proofs and the core contributions of the paper are relegated to the appendix, I question whether a conference publication is the best venue. In conferences, proofs are often reviewed only at a superficial level, and their correctness is not thoroughly verified. It might be more beneficial to submit this work to a theoretical conference or a journal, where reviewers have more time to examine the proofs in depth and provide constructive feedback.

**Essential References Not Discussed:**

I acknowledge that the field of (fair) clustering is vast, making it challenging to cite all relevant references. However, the authors overlook several important works in fair clustering, particularly in the area of representative fairness. Additionally, the citations primarily focus on a specific group of authors, while there are relevant contributions beyond this clique that also deserve recognition. Below, I have listed some relevant references that should be considered to provide a more comprehensive overview of prior research.

[1] Zhang, Zhen, Xiaohong Chen, Limei Liu, Jie Chen, Junyu Huang, and Qilong Feng. "Parameterized Approximation Schemes for Fair-Range Clustering." Advances in Neural Information Processing Systems 37 (2024): 60192-60211.

[2] Gadekar, Ameet, Aristides Gionis, and Suhas Thejaswi. "Fair Clustering for Data Summarization: Improved Approximation Algorithms and Complexity Insights." In proceedings of the ACM Web Conference, 2025.

[3] Thejaswi, Suhas, Ameet Gadekar, Bruno Ordozgoiti, and Michal Osadnik. "Clustering with fair-center representation: Parameterized approximation algorithms and heuristics." In Proceedings of the ACM SIGKDD Conference on Knowledge Discovery and Data Mining, pp. 1749-1759. 2022.

[4] Thejaswi, Suhas, Bruno Ordozgoiti, and Aristides Gionis. "Diversity-aware k-median: Clustering with fair center representation." In Machine Learning and Knowledge Discovery in Databases. Research Track: European Conference, pp. 765-780. Springer, 2021.

[5] Thejaswi, Suhas, Ameet Gadekar, Bruno Ordozgoiti, and Aristides Gionis. "Diversity-aware clustering: Computational Complexity and Approximation Algorithms." arXiv preprint arXiv:2401.05502 (2024).

[6] Chen, Xianrun, Sai Ji, Chenchen Wu, Yicheng Xu, and Yang Yang. "An approximation algorithm for diversity-aware fair k-supplier problem." Theoretical Computer Science 983 (2024): 114305.

**Experimental Designs Or Analyses:**

The experiments are not detailed, as the authors conduct only a single experiment to report the relative cost difference. Additionally, the description does not clearly specify the fairness constraints.

**Methods And Evaluation Criteria:**

Not applicable

**Other Comments Or Suggestions:**

Line 268: it should be $\sigma: S \times C \rightarrow R^+$ right?

**Other Strengths And Weaknesses:**

Given the limited timeframe of the conference review process, it is challenging to provide a thorough review of this paper, as it is theoretically dense and its key contributions are placed in the appendix. The authors should consider reorganizing the paper to include at least some of the key contributions in the main body. If this is not feasible, submitting the work to a journal might be more appropriate, as it would allow for a more detailed review of the proofs. Given these constraints, I cannot vouch for the correctness of the proofs.

**Questions For Authors:**

NA

**Relation To Broader Scientific Literature:**

The paper advances the theoretical understanding of the (fair) clustering problem. The presented approach is non-trivial and requires expertise in the field to develop. While the authors do a commendable job of explaining the method at a high level, I have doubts about its practical applicability to real-world datasets with millions or billions of data points. In my opinion, this work is primarily of theoretical interest.

**Theoretical Claims:**

See my comments in claims and evidences.

---

> ### Author Rebuttal · Authors · 2025-03-31
>
> We thank the reviewer for their thorough and encouraging comments and suggestions, and address their questions below.
>
> ### Re: Experimental description
> The experiment considers the (p,q)-fair k-median problem where each data point is assigned a color of either p or q and each cluster should have a balance of at least p/q. We will include more details in the next version of the paper.
>
> ### Re: Practical applicability
> The run time of our algorithms are $\tilde{O}({nk}+\text{poly}(k/\varepsilon))$. In addition, our algorithm uses uniform sampling, which in contrast to importance sampling is easier to implement at large scale in practice.
> We want to also remark that in our experiments, the bottleneck of the runtime is *not* the construction of the coresets, but rather is to perform fair clustering (which is by the fairtree algorithm by Backurs et al. [ICML’19]). Scaling the algorithm to practical massive scale would require non-trivial engineering efforts, and it is definitely an interesting direction to explore in the future.
>
>
> ### Re: References
> We thank the reviewer for pointing out the literature, and will be sure to include them and provide a more comprehensive overview of prior work in the next version of the paper.
>
> ### Re: Other comments
> Yes, we will make the change in the newer version.
>
> ### Re: Reorganizing to include proofs
> We primarily did not include proofs due to space constraints and instead opted for high level ideas. We will try to incorporate more explanation for lemmas and theorems wherever possible in the newer version.

---

### Official Review · Reviewer_R956 · 2025-03-14

**Overall Recommendation:** 2

**Summary:**

The paper gives coresets for the fair $\(k,z\) $ clustering problem using an oracle model which allows for a combination of queries i) that are returned a weak approximation of distances (weak oracle) at a low cost and ii) queries that are returned exact distances between pair of points but at a high cost (strong oracle). The notion of fairness used here is proportionality-based notion where the proportion of points inside every cluster belonging to a particular demographic group (like gender, income level etc.) must be bounded by some specified thresholds. The idea is to minimize the number of queries to the strong oracle to build coresets i.e. subsample of data that satisfies the assignment criteria with a $(1 \pm \epsilon) $ approximation.

**Claims And Evidence:**

The ideas in the paper are a mix of bunch of ideas from existing coreset literature. for e.g: ring based coresets, weak coresets etc. The techniques are modified to work for the particular version of the clustering problem

**Essential References Not Discussed:**

This is my main issue with this paper. The paper claims ". Prior to Theorem 2, no fair k-median coreset with non-trivial approximation guarantees or coreset size was known."

There are at least 3 papers which discuss coresets for fair clustering
1) Fair Coresets and Streaming Algorithms for Fair k-means, Schmidt et al.
2) On Coresets for Fair Clustering in Metric and Euclidean Spaces and Their Applications, Bandyapadhyay et al.
3) Coresets for clustering with fairness constraints., Huang et al.

1) and 3) are cited but not discussed. I am not sure how the authors can make the above claim when nontrivial coresets do exist. It is not at all clear to me how their coreset results compare/contrast with existing results. This is a big weakness. I believe there should not only be a detailed discussion in terms of quality of coresets, techniques etc with these papers but also, they should be compared with in the experiments section too. These papers have similar techniques for coreset construction (though not in the oracle model). Without both theoretical and experimental comparisons, I am not convinced that this paper has novelty.

**Experimental Designs Or Analyses:**

See the section on methods and evaluation. This is mostly a theoretical paper with just a very small set of experiments. There should have been more comparisons with other sampling techniques.

**Methods And Evaluation Criteria:**

The experimentation section of the paper is rather weak. There is just a baseline comparison with uniform sampling. It is already fairly well known that uniform sampling is unable to give strong relative error guarantees. The authors must have compared to other sampling techniques.

**Other Comments Or Suggestions:**

see response to "Essential References Not Discussed"

**Other Strengths And Weaknesses:**

see responses to other questions

Overall, the paper is not very unclear. However, the presentation could be improved. Experiments should be strengthened and brought to the main body.

**Questions For Authors:**

see response to "Essential References Not Discussed"

**Relation To Broader Scientific Literature:**

The paper proposes coresets for proportionality based fair version of clustering problem. To the best of my knowledge there are some results on coresets for fair clustering and similar ideas like ring based coresets are used. however, I am not sure if there has been a use of a distance oracle model in the existing literature.

**Theoretical Claims:**

There are not many proofs in the main body of the paper. I had a high-level look at the supplementary material for the proofs. They appear ok.

---

> ### Author Rebuttal · Authors · 2025-03-31
>
> We thank the reviewer for their comments and address their questions below.
>
>
> ### Re: Experiments
>
> We chose a uniformly sampled coreset as a baseline since we wanted to compare our algorithm to a method that uses a comparable number of strong oracle queries.
> We do not compare our coreset to other coreset constructions as their trivial application would either require $O(n)$ strong oracle queries which is significantly larger than our $\tilde{O}(k^2 / \varepsilon^2)$ bound or require non-trivial modifications for using less number of strong oracle queries which is out of scope for this work.
>
>
> ### Re: Claim about our result in Theorem 2
>
> Our claim is for the result under the weak-strong oracle model, for which to the best of our knowledge no fair clustering coresets exist and previous results do not extend trivially. We will make this clearer in the next version of the paper. We note that if one wishes to directly apply other algorithms, they would either need to make $O(n)$ strong oracle queries for the trivial application or modify the algorithms to work with fewer queries while providing the $(1+\varepsilon)$ approximation guarantee. For comparison in the classical setting (no weak-strong oracles), our techniques build on top of Braverman et. al (2022) and as a result, assuming access to accurate distances, the bound should be similar. In the next version of the paper, we will be sure to include a more thorough discussion of the techniques of previous papers with comparison to ours.

---

### Decision · Program_Chairs · 2025-05-01

**Decision:**

Accept (poster)

**Comment:**

The paper considers fair clustering problems with inaccurate information: distance information can be obtained from either a strong oracle providing exact distances---but at high cost---and a weak oracle providing potentially inaccurate distance estimates but at low cost. The weak-strong oracle model is motivated by the fact that modern machine learning models often use embedding functions to estimate distances for non-metric data, which can be computationally expensive, and by applications that involve trade-offs between accuracy and cost.  The goal is to produce a near-optimal fair clustering on with a minimum number of strong-oracle queries.  The paper achieves the first (1 + epsilon)-coresets for fair k-median clustering using poly(k, log n, 1/epsilon) queries to the strong oracle. Results are also obtained for coresets without fairness requirements that beat the previous work with high-constant approximations in the fairness-not-considered model.

No previous work considered the fairness model here, and the weak-strong model is well-motivated.